



# Mechanism of Seasonal Arctic Sea Ice Evolution and Arctic Amplification

Kwang-Yul Kim[1*], Benjamin D. Hamlington[2], Hanna Na[3], and Jinju Kim[1]

[1]School of Earth and Environmental Sciences, Seoul National University, Seoul 151-742, Republic of Korea
[2]Department of Ocean, Earth and Atmospheric Sciences, Old Dominion University, Norfork, Virginia 23529, United States of America
[3]Ocean Circulation and Climate Research Center, Korea Institute of Ocean Science and Technology, Ansan, 15627, Republic of Korea

*Correspondence to*: Kwang-Yul Kim (kwang56@snu.ac.kr)

**Abstract.** Sea ice melting is proposed as a primary reason for the Artic amplification, although physical mechanism of the Arctic amplification and its connection with sea ice melting is still in debate. In the present study, monthly ERA-interim reanalysis data are analyzed via cyclostationary empirical orthogonal function analysis to understand the seasonal mechanism of sea ice melting in the Arctic Ocean and the Arctic amplification. While sea ice melting is widespread over much of the perimeter of the Arctic Ocean in summer, sea ice remains to be thin in winter only in the Barents-Kara Seas. Excessive turbulent heat flux through the sea surface exposed to air due to sea ice melting warms the atmospheric column. Warmer air increases the downward longwave radiation and subsequently surface air temperature, which facilitates sea surface remains to be ice free. A 1% reduction in sea ice concentration in winter leads to ~0.76 W m$^{-2}$ increase in upward heat flux, ~0.07 K increase in 850 hPa air temperature, ~0.97 W m$^{-2}$ increase in downward longwave radiation, and ~0.26 K increase in surface air temperature. This positive feedback mechanism is not clearly observed in the Laptev, East Siberian, Chukchi, and Beaufort Seas, since sea ice refreezes in late fall (November) before excessive turbulent heat flux is available for warming the atmospheric column in winter. A detailed seasonal heat budget is presented in order to understand specific differences between the Barents-Kara Seas and Laptev, East Siberian, Chukchi, and Beaufort Seas.





## 1. Introduction

Warming over the Arctic Ocean is observed to accelerate in recent decades. The rate of warming in the Arctic is more than twice the rate of globally averaged warming. This warming and subsequent acceleration is referred to as Arctic amplification (Screen and Simmonds, 2010a, 2010b; Serreze and Barry, 2011). Melting of sea ice in the Arctic Ocean is

suggested to have contributed to the accelerated warming in the lower troposphere (Holland and Bitz, 2003; Serreeze et al., 2009; Kumar et al., 2010; Screen and Simmonds, 2010a, 2010b). The rate of sea ice melting in the Barents and Kara Seas appears to have increased significantly over the last two decades in comparison with that in the earlier period (Stroeve et al., 2007; Comiso et al., 2008; Serreze et al., 2009; Cavalieri and Parkinson, 2012). As should be expected, the accelerated melting of sea ice in the Arctic Ocean has a profound implication on the heat energy budget, sea ice stability, carbon cycle

feedback, and atmospheric and oceanic circulation locally and remotely (IPCC 2013, Serreze and Barry, 2011).

Several physical mechanisms are proposed to explain the accelerated melting of sea ice and warming of lower troposphere. One widely accepted mechanism is the "albedo feedback" (Curry et al., 1995; Screen and Simmonds, 2010a, 2010b; Flanner et al., 2011; Serreze and Barry, 2011). As sea ice melts in the Arctic Ocean, albedo decreases and, as a result, absorption of solar radiation is increased. This certainly is the case in summer when the Arctic sea ice concentration is low and the solar

radiation is highest (Comiso et al., 2008). Also, the nature of air-sea interaction is altered significantly, since sea ice serves as a barrier between the atmosphere and the ocean (Simmonds, 2015). Arctic amplification, on the other hand, is most conspicuous in winter not in summer. Thus, this albedo feedback is considered an indirect cause of sea ice melting and subsequent Arctic amplification in winter.

Another mechanism proposed is the water vapor feedback (Francis and Hunter, 2006; Sedlar et al., 2011; Park et al., 2015).

As warming increases, water vapor content in the atmospheric column increases, leading to an amplified greenhouse effect. Longwave radiation is trapped more in the atmospheric column, resulting in warming of the atmospheric column. In a similar sense, the increased cloudiness due to increased amount of water vapor leaving sea surface may result in an amplification of lower tropospheric warming (Francis and Hunter, 2007).

The most widely accepted mechanism for Arctic amplification is the "insulation feedback". When sea ice remains melted in

winter, turbulent heat flux increases from the open ocean surface, which is instrumental for warming the lower troposphere (Francis et al., 2009; Serreze et al., 2009; Screen and Simmonds, 2010a, 2010b; Deser et al., 2010; Overland et al., 2011; Serreze and Barry, 2011; Cohen et al., 2014; Deser, 2014). According to this hypothesis, increased reception of insolation through the sea surface exposed to air in summer keeps the sea surface warmer and is released in fall and early winter, making the atmosphere warmer. Through this so-called "delayed warming", sea ice remains to be melted in fall and winter,

and excessive turbulent heat flux becomes available through the melted sea ice in winter.

It is not clear, however, why such a mechanism is readily seen only in the Barents and Kara Seas but not in other areas of the Arctic (Petoukhov and Semenov, 2010; Screen and Simonds, 2010b). While summer sea ice melting is clearly seen in areas other than the Barents and Kara Seas, Arctic amplification is observed only in the latter area in winter. Further, the role and





contribution of increased absorption of insolation in summer for increased sea ice melting in winter is not clear, primarily because the region of winter sea ice melting and that of increased insolation reception do not match closely. Thus, it is necessary to understand each term of the feedback process not only from a physical perspective but also in a quantitative one. An accurate quantitative estimation of each term of the feedback process may provide a clearer insight and yield a more

convincing physical mechanism for the feedback process and a reasonable explanation for the regional difference in the Arctic Ocean. Considering the importance of sea ice melting in the overall energy budget and atmospheric and oceanic circulation in the Arctic region, it is also crucial to understand how fast Arctic amplification progresses.

One key issue to be dealt with in the present study is the accurate mechanism of Arctic amplification. Cylostationary empirical orthogonal function (CSEOF) analysis is carried out to identify detailed and physically consistent seasonal

evolution patterns of physical variables associated with sea ice loss in the Arctic Ocean. Specifically, the physical mechanism of sea ice melting and Arctic amplification is investigated from both a spatial and temporal standpoint, so that any delayed response can be explicitly considered. Quantification of each term in the feedback process is attempted in order to clarify the importance of each term in the feedback. Further, the role of water vapor and cloud in the feedback process is assessed. Another key issue to be addressed is why and how sea ice melting in winter develops in the Barents and Kara Seas

but not in the Laptev and Chukchi Seas. This issue is important in order to understand the key components of and reduce uncertainty in the feedback process. Also, it is pivotal to determine how fast the Arctic amplification progresses. The rate of acceleration of the Arctic amplification is estimated based on CSEOF analysis.

## 2. Data and Method of Analysis

The dataset used in the present study is the ERA-interim 1.5°×1.5° monthly reanalysis (Dee et al., 2011) from 1979-2014.

Surface variables analyzed in the present study include sea surface temperature, sea ice concentration, latent and sensible heat fluxes, upward and downward longwave and shortwave radiations, 2 m air temperature. Pressure-level variables analyzed include air temperature, geopotential, zonal wind, meridional wind, and specific humidity. Low-level and total cloud fractions are also analyzed.

The analysis tool employed in this study is the cyclostationary EOF (CSEOF) technique (Kim et al., 1996; Kim and North,

1997). In CSEOF analysis, data $T(r,t)$ are decomposed in the form

$$T(r,t) = \sum_n B_n(r,t)T_n(t), \qquad\qquad (1)$$

where $B_n(r,t)$ are called CSEOF loading vectors (CSLV) and $T_n(t)$ are mutually orthogonal principal component (PC) time series of variable $T(r,t)$ (see section 2a of Kim and North (1997) for details). A primary motivation of CSEOF analysis is to separate the physical component of variability, $B_n(r,t)$, from the stochastic component of variability, $T_n(t)$. Unlike the

EOF loading vector, CSLV is a function of time and describes temporal evolution pertaining to a physical process. Further, CSLV is periodic in time





$$B_n(r,t) = B_n(r,t+d), \qquad (2)$$

where the periodicity $d$ is called the nested period. The nested period represents the periodicity of statistics, which is set to one year in the present study.

A second variable $P(r,t)$ is similarly decomposed into

$$P(r,t) = \sum_n C_n(r,t)P_n(t). \qquad (3)$$

In general, there is no one-to-one correspondence between $\{T_n(t)\}$ and $\{P_n(t)\}$. This means that $\{B_n(r,t)\}$ and $\{C_n(r,t)\}$ are not physically consistent. In order to make physical evolutions derived from two variables to be consistent, $P(r,t)$ should be written as

$$P(r,t) = \sum_n C_n^{(r)}(r,t)T_n(t), \qquad (4)$$

where $C_n^{(r)}(r,t)$ is a new set of loading vectors with corresponding PC time series $\{T_n(t)\}$.

The new set of loading vectors can be determined via the so-called regression analysis in CSEOF space (Kim et al., 2015). It is a two-step process:

$$T_n(t) = \sum_{m=1}^{M} \alpha_m^{(n)} P_m(t) + \varepsilon^{(n)}(t), \qquad (5)$$

and

$$C_n^{(r)}(r,t) = \sum_{m=1}^{M} \alpha_m^{(n)} C_m(r,t), \quad n = 1,2,\cdots, \qquad (6)$$

where $M$ is the number of PC time series used for multivariate regression and $\varepsilon^{(n)}(t)$ is regression error time series. In this study, 20 PC time series are used for regression ($M = 20$). As a result of regression analysis in CSEOF space, entire data (variables) can be written as

$$Data(r,t) = \sum_n \left\{ B_n(r,t), C_n^{(r)}(r,t), D_n^{(r)}(r,t), E_n^{(r)}(r,t), \cdots \right\} T_n(t), \qquad (7)$$

where the terms in curly braces represent physically consistent evolutions derived from different variables. As should be clear from (7), a primary motivation of CSEOF analysis is to understand details of physical processes by extracting evolutions from various atmospheric and oceanic variables in a physically consistent manner. The variable $T(r,t)$ is called the target variable and is determined in such a way that the physical process under investigation is clearly identified and separated as a single CSEOF mode.





## 3. Results and Discussion

Northern Hemispheric (30°-90° N) 2 m air temperature is used as the target variable, since polar amplification in the Northern Hemisphere is clearly identified as the leading mode aside from the seasonal cycle. Then, CSEOF analysis followed by regression analysis is conducted on all other (predictor) variables to extract physically consistent evolutions from these variables. Table 1 shows the $R^2$ values of regression for different variables.

### 3.1 Seasonal patterns of sea ice concentration

Figure 1 shows the average seasonal patterns of sea ice concentration in the Arctic Ocean. The sea ice boundary in the Atlantic sector appears to be most volatile throughout the year. In the Russian and Canadian sectors of the Arctic Ocean, the ice boundary abuts the continents in winter and spring, but retreats to the north in summer and fall. During the melting season, sea ice concentration decreases significantly in the Laptev, East Siberian, Chukchi and Beaufort Seas.

### 3.2 The warming mode and associated anomalous patterns

Figure 2 shows the first CSEOF mode of surface (2 m) air temperature (SAT); it explains ~15 % of the total variability. This mode is well separated dynamically from the second CSEOF mode, which represents Arctic oscillation; its PC time series is correlated at 0.67 with the ±12-month moving averaged AO index. For the sake of brevity, seasonally averaged patterns of the CSLV are presented instead of monthly patterns. Both the CSLV and the corresponding PC time series clearly show that this mode represents warming in the Northern Hemisphere. In particular, the PC time series shows a conspicuous trend during the study period, indicating a persistent increase in SAT. Seasonal variation of the pattern and magnitude of warming is clear with significant warming in winter and weak warming in summer. Other striking features include pronounced warming over the Barents-Kara Seas in winter and weak cooling in East Asian mid-latitudes (see also Fig. S2). According to the PC time series, an acceleration of warming is obvious in the Arctic region, particularly over the Barents-Kara Seas. In particular, 2006/07 warming in winter seems to have been unprecedented (Stroeve et al., 2008; Kumar et al., 2010).

Figure 3 shows the regressed seasonal patterns of sea ice concentration and radiation anomalies corresponding to the warming mode shown in Fig. 2. The anomalous pattern of sea ice concentration in winter looks similar to that in spring. On the other hand, the summer pattern looks similar to that in fall. In winter and spring, conspicuous melting of sea ice is primarily in the Barents-Kara Seas, whereas sea ice melting is widespread in the Laptev, East Siberian, Chukchi, and Beaufort Seas in summer and fall.

In winter, when insolation is weak, net longwave radiation is upward over the region of sea ice melting, while it is downward over much of the Arctic Ocean, particularly in the Atlantic sector. As sea ice melts, warmer sea surface is exposed to air yielding increased upward longwave radiation in the Barents-Kara Seas. In the North Atlantic Ocean, where sea ice





concentration is already low (Fig. 1), net longwave radiation is downward, suggesting that increase in atmospheric temperature is larger than that of sea surface temperature. In late spring (May), downward shortwave radiation increase significantly over the region of sea ice loss. The increase in shortwave radiation is much larger than the net longwave radiation, thereby resulting in net downward radiation flux over the region of sea ice loss. In summer, sea ice melting

expands into the Laptev, Chukchi, and Beaufort Seas. There is little change in net longwave radiation, but downward shortwave radiation increases significantly over the region of sea ice loss. This marked increase in downward shortwave radiation in spring and summer is associated with the decreased albedo as sea ice melts. In fall, the anomalous pattern of sea ice concentration is similar to that in summer, but the change in net longwave and shortwave radiation is small.

Figure 4 shows the seasonal patterns of anomalous sensible and latent heat fluxes. In winter, sensible heat flux and, to a

lesser extent, latent heat flux increases over the Barents and Kara Seas. Over the North Atlantic the anomalous surface flux is downward, primarily because of the increased atmospheric temperature; heat flux is reduced, since the difference between sea surface temperature and air temperature is reduced due to atmospheric warming. In spring, a similar increase in turbulent heat flux is clearly seen over the Barents-Kara Seas. In summer, there is little change in turbulent heat flux although the area of sea ice melting is much expanded (Simmonds and Rudeva, 2012); note that there is little change in air

temperature in summer (Fig. 2c). In fall, turbulent heat flux is increased primarily in the Kara and Chukchi Seas because a wider area of sea surface is exposed to colder air above.

Figure 5 shows the seasonal patterns of anomalous net radiation and turbulent heat flux. In spring, net downward radiation and upward heat flux are similar in magnitude. In summer, there is net downward radiation, which derives primarily from the increased absorption of solar radiation owing to decreased albedo (Serreze and Francis, 2006; Serreze et al., 2009; Screen

and Simmonds, 2010a). In fall heat flux is increased over the region of sea ice loss, but the amount of heat flux released is much less than the increased amount of shortwave radiation absorbed in summer. In winter, a significant increase in turbulent heat flux is observed over the Barents-Kara Seas and a reduction of turbulent heat flux in the North Atlantic

### 3.3 Seasonal patterns of sea surface temperature

While sea surface temperature is observed to increase over the region of sea ice loss in summer and fall, anomalous sea surface temperature vanishes in the Laptev, East Siberian, Chukchi, and Beaufort Seas as sea ice recovers over the area (Fig. 6). It should be pointed out that the increased net downward radiation in summer, and henceforth the increased sea surface temperature in summer and fall, does not lead to a pronounced thinning of sea ice in winter (see Fig. 6a). Instead, sea ice melting is confined to the Barents-Kara Seas in winter, where turbulent heat flux is significantly increased. It seems that the

increased solar radiation as a result of albedo feedback is responsible for the sea ice loss and sea surface warming in summer, except for the western part of the Barents Sea, where sea surface warming seems associated with oceanic heat transport. The



increased energy, however, does not seem connected, at least directly, with the increased turbulent heat flux in winter. Note that the region of sea surface warming in summer does not match well with the region of sea ice loss in winter (Fig. 6).

**3.4 Mechanism of sea ice melting**

While significant melting is observed only during summer and fall over the Laptev and Chukchi Seas, sea ice melting continues throughout the year over the Barents-Kara Seas (see different regions of conspicuous sea ice melting in Fig. 5). In order to understand why sea ice distribution differs markedly over the Barents and Kara Seas, the monthly energy budget is computed in Fig. 7a. In April-June, absorption of shortwave radiation increases dramatically over the region; this excessive incoming energy explains the bulk of the total energy budget. During the rest of the year, net radiation change is fairly small

($< 3$ W m$^{-2}$). On the other hand, turbulent energy is released mainly during January-April in addition to November when air temperature becomes much colder than sea surface temperature. The total incoming energy seems to be nearly in balance with the total outgoing energy.

As shown in Fig. 7b, the variation of the SAT over the Barents-Kara Seas is highly consistent with those of the downward longwave radiation (corr=0.965) and the upward longwave radiation (corr=0.991). Figure 7c shows that the monthly

variation of the 850 hPa air temperature is more strongly correlated with the downward longwave radiation (corr=0.856) than the upward longwave radiation (corr=0.707). It appears that the lower tropospheric temperature essentially determines the strength of the downward longwave radiation. The upward longwave radiation is determined primarily by the SAT. It should be noted that the net longwave radiation is upward in late fall-early spring (Nov-May). It is, then, immediately obvious that SAT cannot increase continuously in the absence of any other energy flux. As a result, this process cannot be

sustained without any additional source of energy.

It is noted that both the downward and upward radiation at the surface is maximized in winter (specifically February) with very small values in summer (Fig. 7b). It is also worthy of remark that turbulent heat flux is maximized when 850 hPa temperature is minimum in March and November (Fig. 7c). The energy budget in the Barents and Kara Seas indicates that the release of turbulent flux through the sea surface exposed to air is a major component of energy source in winter (Fig. 7a).

It appears that sea ice melting condition persists in winter, so that turbulent heat flux released from the surface of the ocean reaches a maximum in March.

This physical relationship between temperature and longwave radiation differs significantly in the Laptev or Chukchi Seas, where net upward longwave radiation is maximized in October (Figs. 8a and 8b). Further, the energy budget exhibits substantially different seasonal patterns with a significant upward energy flux only briefly in October. Both the net radiation

and turbulent heat flux contribute to this net upward energy flux in October, which is smaller in magnitude than that in the Barents-Kara Seas. The most striking difference is the magnitude of turbulent heat flux in January-April. Turbulent heat flux in January-April is much smaller in the Laptev and Chukchi Seas than in the Barents and Kara Seas. Thus, it seems that





the increased absorption of shortwave via ice-albedo feedback in summer and the resulting delayed warming is not so effective in maintaining sea ice stay melted in winter in the Laptev and Chukchi Seas.

It is noted that the magnitude of the net longwave radiation in late fall (October or November) is generally smaller than that of net turbulent heat flux in all three sea-ice melting regions studied here (Deser et al., 2010; Screen et al., 2013). This result is not entirely consistent with the conclusion in earlier studies (see Serreze et al., 2009) that heat energy stored in summer is released in the form of longwave radiation in cold seasons. It is clear that the magnitude of "delayed warming" (delayed release of energy from the ocean to the atmosphere) is much less than the increased absorption of insolation at sea surface during summer (Figs. 8a and 8b). It is not clear based on data analysis alone whether this excessive energy is transported to other regions in the Arctic Ocean or is sequestered into the depth of the ocean.

Such a distinct behavior can be understood in terms of the distinct evolution of sea ice concentration in the three regions. Figure 8c shows that sea ice melting is maximized in July-October in the Laptev or Chukchi Seas. By November, sea ice refreezes and sea ice concentration becomes nearly normal. Therefore, the release of turbulent heat flux through the exposed sea surface quickly diminishes to zero. Further, relatively warm air in August-October prevents vigorous release of turbulent heat flux through the exposed sea surface. On the other hand, sea ice melting remains significant throughout late fall and winter in the Barents and Kara Seas, which provides a favorable condition for releasing turbulent heat flux through the exposed sea surface.

### 3.5 Arctic Amplification

While net longwave radiation is generally small compared to other energy terms throughout the year, it is an essential ingredient for sea ice melting and subsequent atmospheric warming. Although the net longwave radiation is less than 3 W m$^{-2}$ (Fig. 7a), upward and downward component of longwave radiation individually reach maximum values of ~15 W m$^{-2}$ in February (Fig. 7b), which is larger than the maximum turbulent flux in March. On the other hand, the upward longwave radiation is, in general, larger than the downward longwave radiation, resulting in a net deficit of longwave radiation at surface, which is not a favorable condition for maintaining ice-free condition; sea ice loss due to increased downward longwave radiation is followed by sea ice gain due to increased upward longwave radiation. Therefore, longwave radiation, by itself, cannot explain the winter loss of sea ice in the Barents-Kara Seas unless other mechanisms are invoked. It is the release of turbulent heat flux through the exposed sea surface, which facilitates the open sea surface to survive cold winter without refreezing. The turbulent heat flux warms the lower troposphere and increases the downward longwave radiation.

Figure 9 shows the schematic of the arctic amplification feedback process and the budgetary numbers derived from the January-March average pattern of the loading vector. A 1 % reduction in sea ice concentration leads to ~0.76 W m$^{-2}$ increase in upward heat flux, which in turn increases 850 hPa temperatures by ~0.07 K (see upper panels of Fig. S2). Tropospheric warming leads to an increase in downward longwave radiation by ~0.97 W m$^{-2}$, which yields ~0.26 K increase in 2 m temperature.



This mechanism, in principle, is essentially identical with that proposed by Screen and Simmonds (2010a, 2010b) and Serreze et al. (2009).  It should be noted, however, that excessive absorption of insolation during summer is not a necessary and sufficient condition for the positive feedback process.  While sea ice melting is significant and absorption of insolation is clearly reflected in the warming of sea surface in summer (Figs. 5 and 6), and, as a result, atmospheric temperature is

warmer in autumn (Fig. 2), no feedback process is developed in winter over the Laptev or Chukchi Seas; sea ice refreezes in fall as atmospheric temperature drops much below freezing.  In the Barents-Kara Seas, shortwave radiation absorbed during summer may help facilitate the feedback process discussed here.  On the other hand, absorbed shortwave radiation in summer may not necessarily be a unique contributor to the feedback process.  For example, heat transport by the warm Norwegian current may prevent melted sea ice from refreezing in fall and winter (Chylek et al., 2009; Årthun et al., 2012;

Onarheim et al., 2015).  Årthun et al. (2012), Smedsrud et al. (2013), and Onarheim et al. (2015) showed that there is a substantial link between the ocean heat transport into the western Barents Sea and the sea ice variability in the Barents-Kara Seas.  The DJF (December-January-February) pattern of sea surface temperature anomaly in Fig. 6 supports their analysis.  It is clear, however, that oceanic heat transport alone cannot explain all the major features of sea ice reduction in the Barents-Kara Seas.  It should be pointed out that the magnitude of sea surface warming is much smaller than that of atmospheric

warming (see Figures 2 and 6).

As shown in Fig. 10, the anomalous patterns of surface air temperature, longwave radiation, and turbulent flux are closely related to that of sea ice reduction.  The winter pattern of specific humidity (see also supplementary Fig. S1) is also highly correlated with that of 850 hPa temperature (pattern corr=0.88) and of downward longwave radiation (pattern corr=0.81).  In the Barents and Kara Seas, the magnitude of winter specific humidity increases by 0.037 g kg$^{-1}$ per 1 % reduction in sea ice

concentration.  It appears that the increased atmospheric temperature is responsible for the increased specific humidity.  In turn, the increased specific humidity may have contributed to an increase in atmospheric temperature by trapping more longwave radiation (Francis and Hunter, 2007; Screen and Simmonds, 2010a).  Thus, the increase in specific humidity together with the increase in atmospheric temperature may result in increased downward longwave radiation.  The winter pattern of total cloud cover, however, is not significantly correlated with that of downward longwave radiation (see Fig. S1).

Thus, it does not seem likely that change in cloud cover is responsible for the increased downward longwave radiation (Screen and Simmonds, 2010b) in the Barents and Kara Seas; this finding is somewhat different from that of Schweiger et al. (2008).

According to the PC time series in Fig. 2b, this positive feedback process is accelerating in time.  The rate of acceleration can be estimated from the PC time series.  Let us consider an exponential fit to the PC time series in the form

$$T(t) = a\exp(\gamma t) + b = a(e^{\gamma})^{t} + b \triangleq a(1+\gamma)^{t} + b, \tag{8}$$

where $t$ is time in years since 1979.  A least square fit yields $a = 0.2$, $b = -1.0$, and $\gamma = 0.08$ (see blue dashed curve in Fig. 2b).  Thus, sea ice melting accelerates at the rate of ~8 % annually.  Since the present winter sea ice concentration in the Barents-Kara Seas is ~40 %, sea ice melting will increase by ~4.8 % (=60 % × 0.08) next year.  This sea-ice reduction rate is



higher than other studies, which predict sea ice disappearance by mid-to-end of this century (Stroeve et al., 2007; Serreze et al., 2007; Boé et al., 2009a; Wang and Overland, 2009). Earlier studies, however, are not specific about the sea ice in the Barents-Kara Seas. Also, uncertainty is inherent in model projections, since most climate models do not accurately simulate the complex Arctic feedbacks (Boé et al., 2009b; English et al., 2015). Uncertainty is obvious in our estimate, since it is

5 based on the exponential curve fitting, which is an important caveat; the result should be understood accordingly.



## 4. Concluding Remarks

CSEOF analysis was conducted to investigate the physical mechanism of sea ice melting in the Arctic Ocean and the Arctic amplification. The Arctic warming mode was extracted from Northern Hemispheric (30°-90° N) surface (2 m) air temperature, which clearly depicts the amplification pattern in the Arctic. Then, regression in CSEOF space was conducted

on all other variables to understand the concerted variation of various climate variables involved in the physical mechanism of sea ice melting and Arctic amplification.

While sea ice reduction occurs over much of the perimeter of the Arctic Ocean, ice-free condition persists in winter only in the Barents-Kara Seas. The primary reason is that the release of turbulent heat flux from the exposed sea surface in winter is currently possible only over the Barents-Kara Seas (see Fig. 10e). Over the other ocean basins including the Laptev and

Chukchi Seas, sea ice refreezes quickly in late fall and closes up the exposed sea surface; as a result, excessive turbulent heat flux is not available in winter in these ocean basins.

Our analysis confirms that the temporal pattern of sea ice variation indeed differs significantly between the Barents-Kara Seas and the Laptev and Chukchi Seas. Sea ice refreezes and the sea surface exposed to air is closed up in late fall in the Laptev and Chukchi Seas. As a result, significant absorption of solar radiation in summer does not lead to increased

turbulent heat flux in winter. On the other hand, sea surface does not freeze up completely in the Barents-Kara Seas. Consequently, turbulent heat flux becomes available in winter in the Barents-Kara Seas for heating the atmospheric column (Fig. 10f), which in turn increases downward longwave radiation (Fig. 10d). The delayed warming from summer energy absorption via albedo feedback (Screen and Simmonds, 2010a; Serreze and Barry, 2011) does not appear to be a necessary and sufficient condition for the feedback process; it appears that the delayed warming is not uniquely responsible for

prolonged sea ice melting in the Barents-Kara Seas; for example, increased ocean heat transport into the western Barents Sea may have provided a favorable condition for the sustenance of ice-free sea surface in winter. Wind may also be partially responsible for sea ice reduction (Ogi and Wallace, 2012).

The increased insolation in spring and summer decreases sea ice concentration along the perimeter of the Arctic Ocean. This thinning of sea ice, in turn, increases the absorption of solar radiation at the exposed ocean surface. There is, however, no

direct indication that the absorbed insolation is later used to keep the sea surface remain ice free in winter, although the warmer sea surface may have delayed sea ice refreezing. In the Laptev, East Siberian, and Chukchi Seas, upward longwave radiation and heat flux increase briefly in October, and sea ice refreezes in November, suggesting that sea surface warming in summer and fall has not sufficiently delayed sea ice refreezing. Therefore, the increased absorption of insolation does not contribute, at least directly, to the melting of sea ice in winter in these ocean basins. In the Barents-Kara Seas, upward

radiation and heat flux increase briefly in November, and then decrease in December. Unlike the other areas, however, sea surface remains to be exposed to cold air and turbulent heat flux increases significantly in January-March in the Barents-Kara Seas (Fig. 10a). Again, there is no concrete evidence that the absorbed insolation in summer is used directly in the melting of sea ice in winter.




In the Barents and Kara Seas, a 1 % reduction in sea ice concentration in winter leads to ~0.76 W m$^{-2}$ increase in upward heat flux. This excessive flux is used to warm the lower troposphere; 850 hPa air temperature increases by ~0.07 K. The raised air temperature, in turn, increases downward longwave radiation by ~0.97 W m$^{-2}$. As a result, surface air temperature increases by ~0.26 K, which helps maintain the ice-free condition (see also Fig. 10). Such a mechanism persists throughout

the winter, since sea ice does not refreeze, at least completely, until turbulent heat flux is sufficiently increased during cold winter. Specific humidity increases as atmospheric temperature increases; the anomalous patterns of the two are highly correlated. Thus, it appears that the increased specific humidity may have also contributed to the increase in downward longwave radiation. The anomalous pattern of cloud cover, however, is not significantly correlated with that of atmospheric temperature, suggesting that change in cloud cover has not significantly contributed to the Arctic amplification.

The physical process of sea ice melting and increased air temperature appears to have been accelerating. According to a simple exponential fitting to the PC time series of the warming mode, the strength of this positive feedback process increases by ~8 % every year. At this rate, surface air temperature (850 hPa temperature) may increase by ~10 K (~3 K) over the Barents and Kara Seas with respect to the 1979 winter mean value as sea ice completely melts (see also IPCC, 2013).

It should be pointed out that several different mechanisms have been invoked to explain Arctic amplification. For example,

Hall (2004), Alexeev et al. (2005), Graversen and Wang (2009), and Graversen et al. (2014) showed based on model experiments that surface albedo feedback explains a large fraction of polar temperature amplification. Pithan and Mauritsen (2014) and Graversen et al. (2014) demonstrated that lapse-rate feedback also contributes to polar amplification using climate models. Finally, it should be pointed out that there are different mechanisms by which atmospheric moisture can be transported to the Barents and Kara Seas. For example, Sorteberg and Walsh (2008) demonstrated that moisture transport

into the Arctic has increased due to increased seasonal cyclonic activity. Simmonds and Keay (2009) showed that the trends and variability in September ice coverage is related to the mean cyclone characteristics. Park et al. (2015) showed that downward infrared radiation in the Arctic is driven by horizontal atmospheric water flux and warm air advection into the Arctic. Simmonds and Govekar (2014) also argued that sea ice reduction in the Arctic may be due to the advection of warm and humid air into the Arctic. In light of different views on sea ice loss and temperature amplification in the Arctic, the

present study should be understood as a contrasting and complementary view on the mechanism of sea ice loss and temperature amplification in the Arctic.

*Data Availability*: All the results of analysis in the present paper are freely available by contacting the corresponding author.

*Acknowledgments*: This research was supported by SNU-Yonsei Research Cooperation Program through Seoul National University in 2015.



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



**Table Captions**

**Table 1**. Variables used in the present study with units and $R^2$ values of regression. The target variable for regression is 2 m air temperature.

5   **Figure Captions**

**Figure 1**. Geography of the Arctic Ocean (69°-90° N) and the seasonal patterns of average sea ice concentration (%) based on 1979-2014 ERA interim data.

**Figure 2**. The seasonal patterns of the Northern Hemispheric (30°-90° N) warming mode (upper panel; 0.3 K) and the corresponding amplitude time series (lower panel). The dashed curve is an exponential fit (see Eq. (8)) to the PC time series.

10   **Figure 3**. The regressed seasonal patterns of sea ice concentration (shading; 1 %), net shortwave radiation (red contours; 1 W m$^{-2}$), and net longwave radiation (black contours; 0.5 W m$^{-2}$) in the Arctic region (64.5°-90° N). Net upward longwave radiation and net downward shortwave radiation are defined as positive.

**Figure 4**. The regressed seasonal patterns of sea ice concentration (shading; 1 %), sensible heat flux (red contours; 1 W m$^{-2}$), and latent heat flux (black contours; 0.2 W m$^{-2}$) in the Arctic region (64.5°-90° N). Net upward heat flux is defined as 15   positive.

**Figure 5**. The regressed seasonal patterns of sea ice concentration (shading; 1 %) and net surface radiation (black contours; 2 W m$^{-2}$), and turbulent heat flux (red contours; 2 W m$^{-2}$) in the Arctic region (64.5°-90° N). Positive values represent upward radiations and heat fluxes. The three green boxes represent the regions of significant change in sea ice concentration: Barents and Kara Seas [21°-79.5° E × 75°-79.5° N], Laptev Sea [105°-154.5° E × 76.5°-81° N], and Chukchi 20   Sea [165°-210° E × 72°-76.5° N].

**Figure 6**. The regressed seasonal patterns of sea surface temperature (shading; 0.05 K) and the reduction of sea ice concentration (contours; 2 %) in the Arctic region (64.5°-90° N).

**Figure 7**. (a) Monthly values of total energy flux (black), net longwave radiation (red dotted), net shortwave radiation (red dashed), net radiation (red solid), latent heat flux (blue dotted), sensible heat flux (blue dashed), and turbulent heat flux (blue 25   solid) in the Barents-Kara Seas (21°-79.5° E × 75°-79.5° N). (b) Monthly plot of 2 m air temperature (black), downward longwave radiation (red), and upward longwave radiation (blue). (c) Monthly plot of 850 hPa air temperature (black), downward longwave radiation (red), and upward longwave radiation (blue).

**Figure 8**. Monthly values of total energy flux (black), net longwave radiation (red dotted), net shortwave radiation (red dashed), net radiation (red solid), latent heat flux (blue dotted), sensible heat flux (blue dashed), and turbulent heat flux (blue 30   solid) in the (a) Laptev Sea [105°-154.5° E × 76.5°-81° N], and (b) Chukchi Sea [165°-210° E × 72°-76.5° N]. (c) Monthly sea ice concentration change in the Barents-Kara Seas (red), Laptev Sea (blue), and Chukchi Sea (black).



**Figure 9**. Sea ice melting mechanism in the Barents and Kara Seas. The numbers represent the magnitude of change at each step for a 1 % reduction in sea ice concentration and are based on the January-March averaged loading vector of the warming mode.

**Figure 10**. The regressed DJF patterns of (a) sea ice (shading) and 2 m air temperature (contour), (b) 900 hPa specific
5   humidity, (c) upward longwave radiation at surface, (d) downward longwave radiation at surface, (e) turbulent (sensible + latent) heat flux, and (f) 850 hPa air temperature. The green contours in (b)-(f) represent sea ice concentration in (a).



**Table 1**. Variables used in the present study with units and $R^2$ values of regression. The target variable for regression is 2 m air temperature.

| Variable | $R^2$ value |
|---|---|
| sea ice (fraction) | 0.960 |
| sea surface temperature (°C) | 0.937 |
| downward longwave radiation (W m$^{-2}$) | 0.995 |
| upward longwave radiation (W m$^{-2}$) | 0.999 |
| net shortwave radiation (W m$^{-2}$) | 0.907 |
| sensible heat flux (W m$^{-2}$) | 0.968 |
| latent heat flux (W m$^{-2}$) | 0.954 |
| low cloud cover (fraction) | 0.947 |
| total cloud cover (fraction) | 0.921 |
| specific humidity (g kg$^{-1}$) | 0.945 |
| air temperature (1000-850 hPa; °C) | 0.962 |
| geopotential (1000-850 hPa; m$^2$ s$^{-2}$) | 0.772 |
| wind (1000-850 hPa; m s$^{-1}$) | 0.844 |





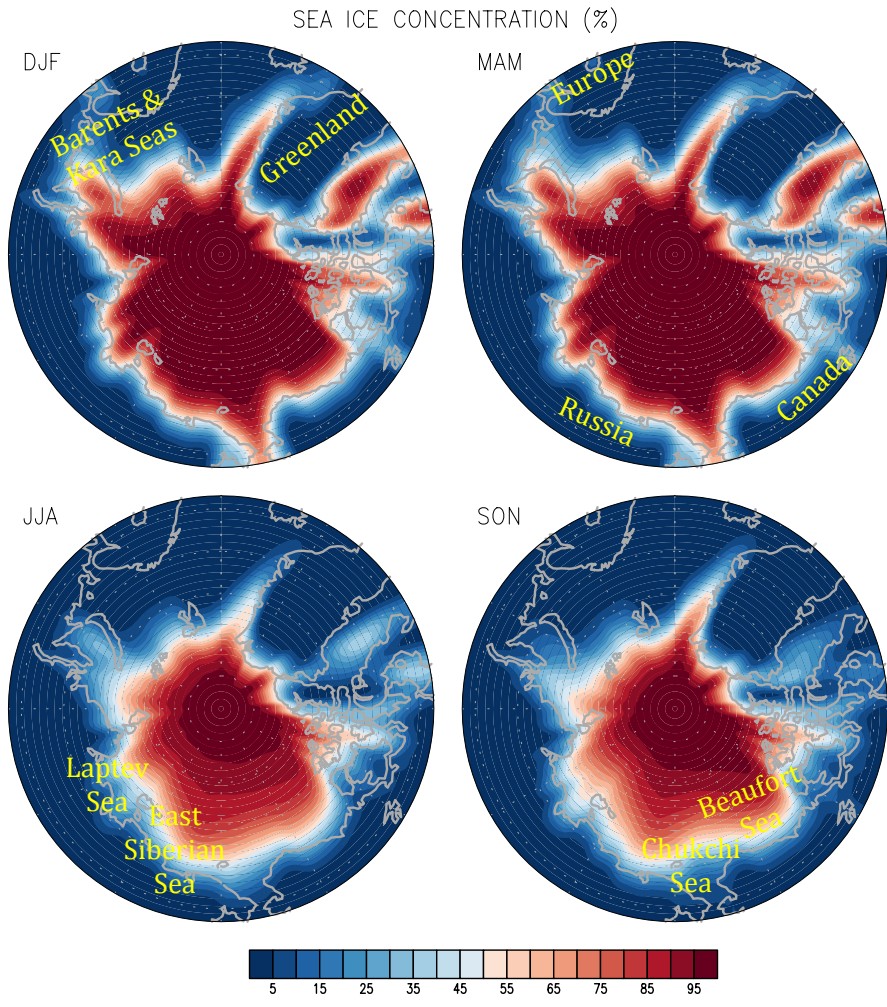

**Figure 1**. Geography of the Arctic Ocean (69°-90° N) and the seasonal patterns of average sea ice concentration (%) based
on 1979-2014 ERA interim data.





**Figure 2**. The seasonal patterns of the Northern Hemispheric (30°-90° N) warming mode (upper panel; 0.3 K) and the corresponding amplitude time series (lower panel). The dashed curve is an exponential fit (see Eq. (8)) to the PC time series.





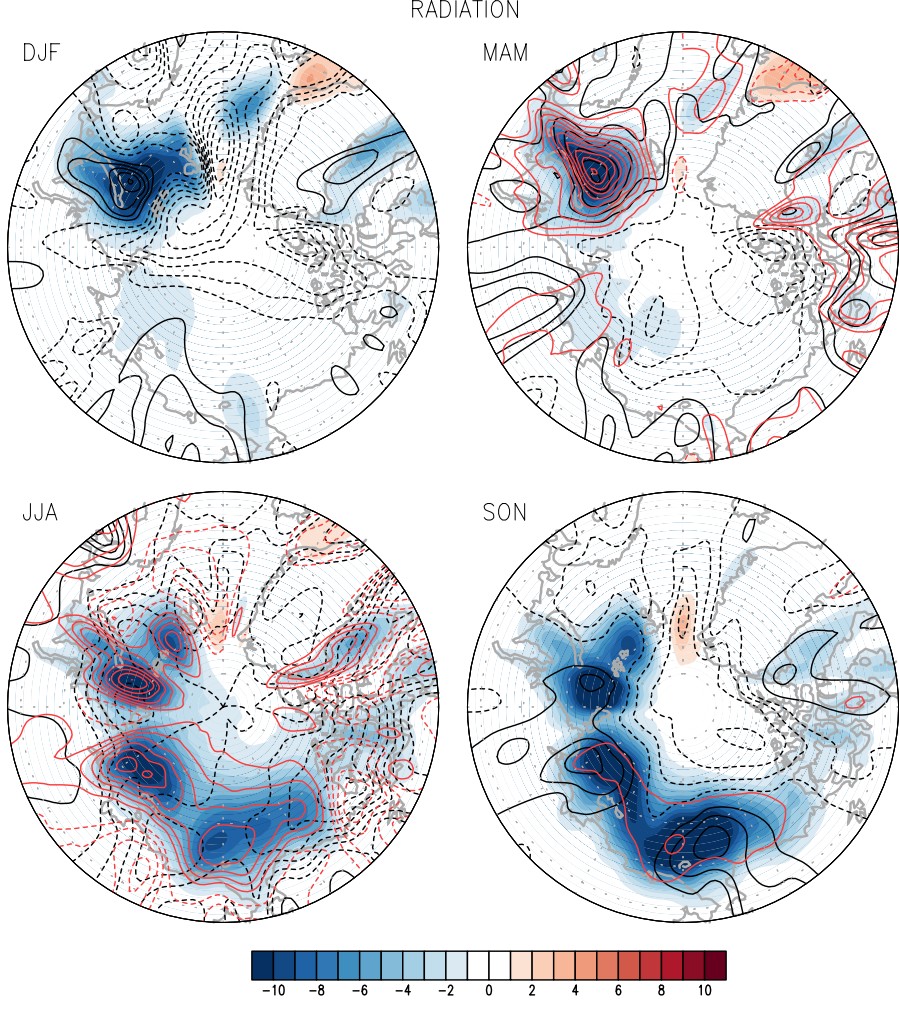

**Figure 3**. The regressed seasonal patterns of sea ice concentration (shading; 1 %), net shortwave radiation (red contours; 1 W m$^{-2}$), and net longwave radiation (black contours; 0.5 W m$^{-2}$) in the Arctic region (64.5°-90° N). Net upward longwave radiation and net downward shortwave radiation are defined as positive. Solid contours represent positive values and dotted contours represent negative values.





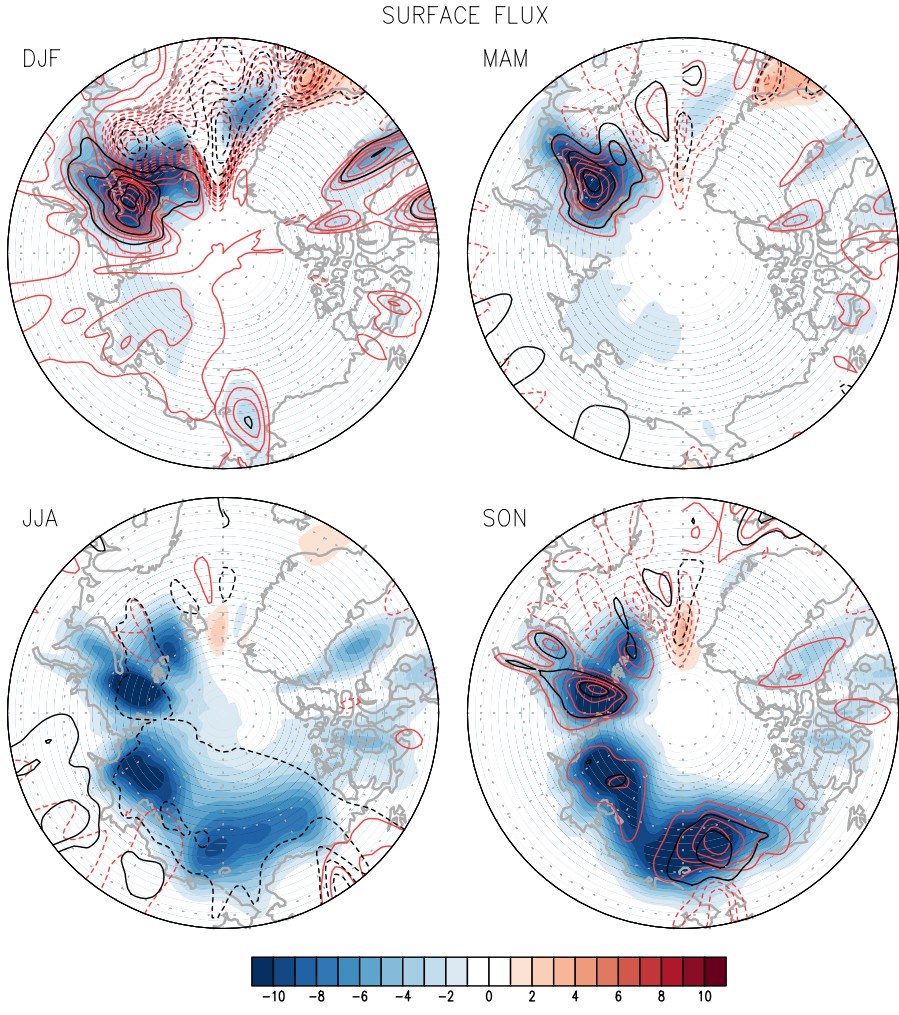

**Figure 4**. The regressed seasonal patterns of sea ice concentration (shading; 1 %), sensible heat flux (red contours; 1 W m$^{-2}$), and latent heat flux (black contours; 0.2 W m$^{-2}$) in the Arctic region (64.5°-90° N). Net upward heat flux is defined as positive.



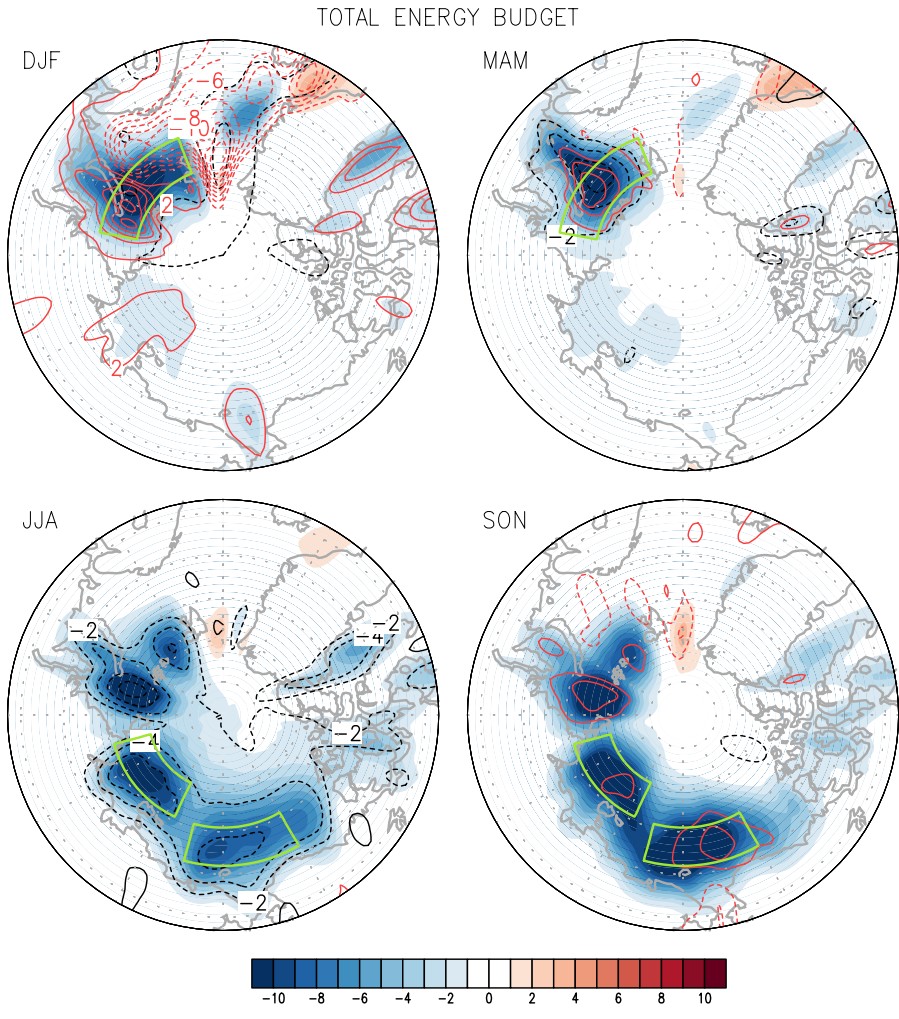

**Figure 5**. The regressed seasonal patterns of sea ice concentration (shading; 1 %) and net surface radiation (black contours; 2 W m$^{-2}$), and turbulent heat flux (red contours; 2 W m$^{-2}$) in the Arctic region (64.5°-90° N). Positive values represent upward radiations and heat fluxes. The three green boxes represent the regions of significant change in sea ice concentration: Barents and Kara Seas [21°-79.5° E × 75°-79.5° N], Laptev Sea [105°-154.5° E × 76.5°-81° N], and Chukchi Sea [165°-210° E × 72°-76.5° N].





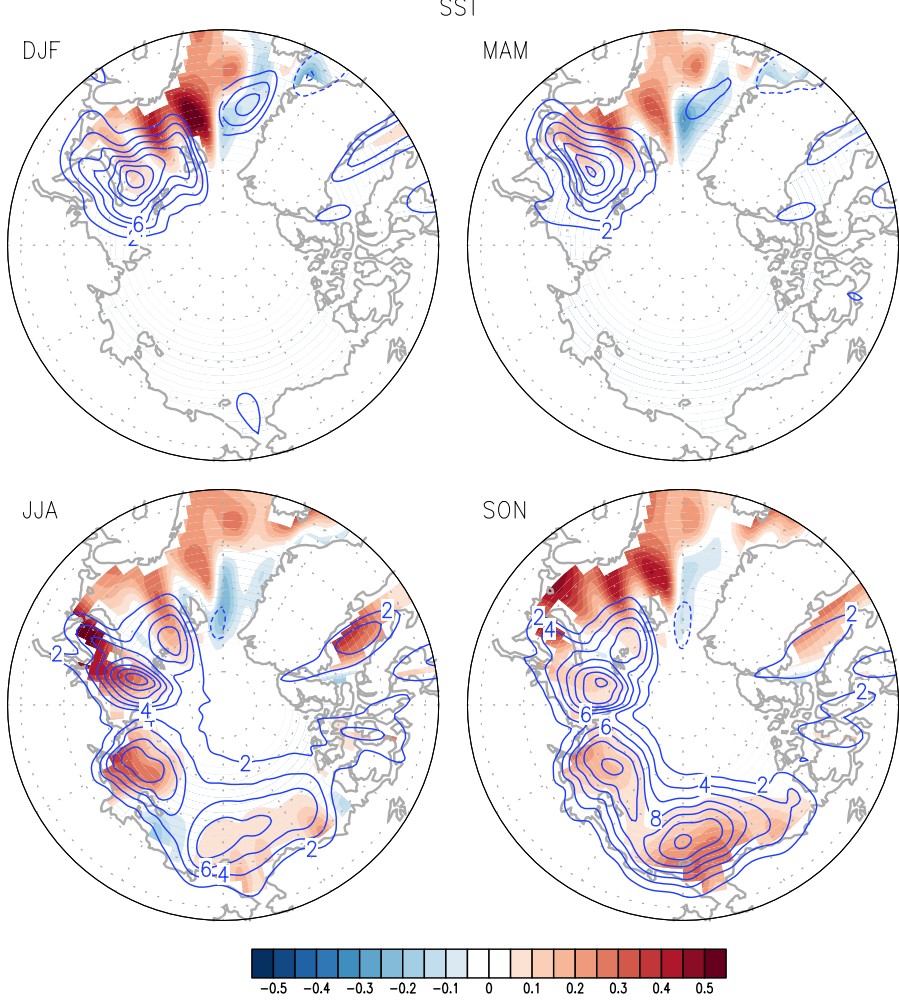

**Figure 6**. The regressed seasonal patterns of sea surface temperature (shading; 0.05 K) and the reduction of sea ice concentration (contours; 2 %) in the Arctic region (64.5°-90° N).



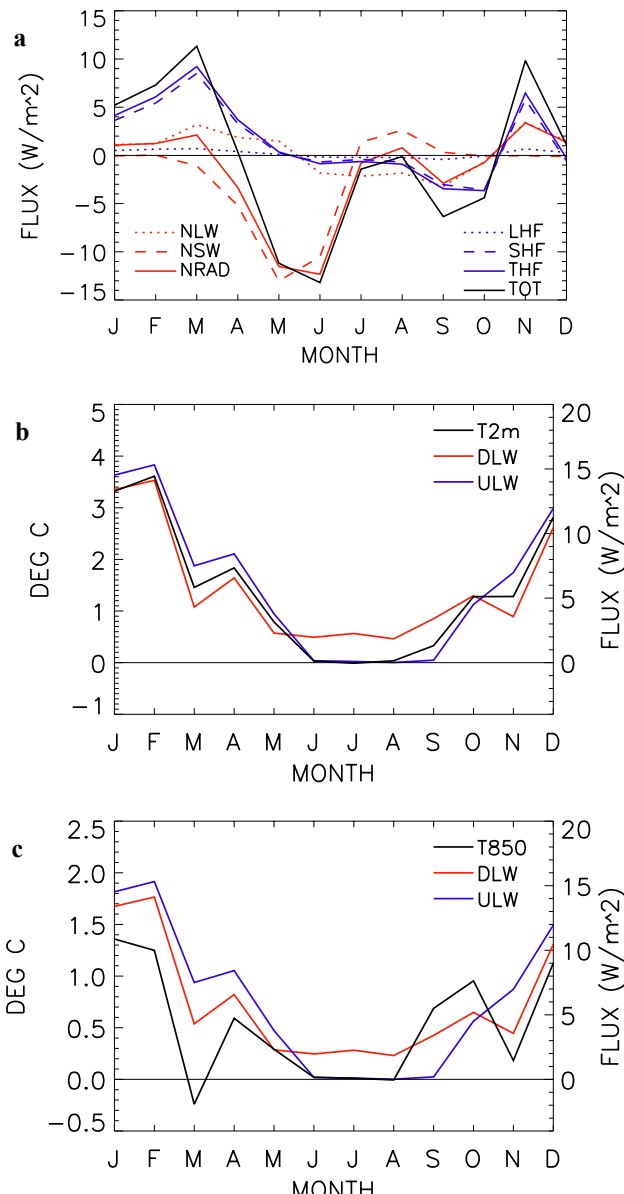

**Figure 7**. (a) Monthly values of total energy flux (black), net longwave radiation (red dotted), net shortwave radiation (red dashed), net radiation (red solid), latent heat flux (blue dotted), sensible heat flux (blue dashed), and turbulent heat flux (blue solid) in the Barents-Kara Seas (21°-79.5° E × 75°-79.5° N). (b) Monthly plot of 2 m air temperature (black), downward longwave radiation (red), and upward longwave radiation (blue). (c) Monthly plot of 850 hPa air temperature (black), downward longwave radiation (red), and upward longwave radiation (blue).





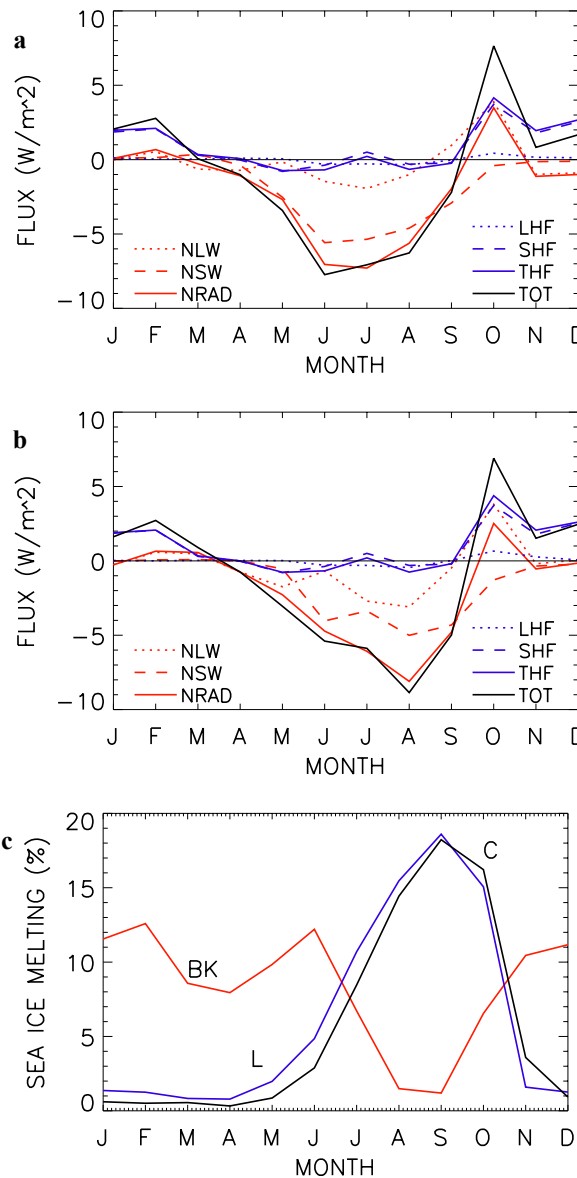

**Figure 8**. Monthly values of total energy flux (black), net longwave radiation (red dotted), net shortwave radiation (red dashed), net radiation (red solid), latent heat flux (blue dotted), sensible heat flux (blue dashed), and turbulent heat flux (blue solid) in the (a) Laptev Sea [105°-154.5° E × 76.5°-81° N], and (b) Chukchi Sea [165°-210° E × 72°-76.5° N]. (c) Monthly
10   sea ice concentration change in the Barents-Kara Seas (red), Laptev Sea (blue), and Chukchi Sea (black).



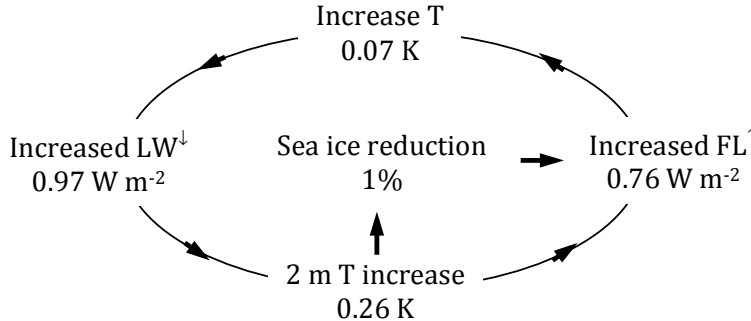

**Figure 9**. Sea ice melting mechanism in the Barents and Kara Seas. The numbers represent the magnitude of change at each step for a 1 % reduction in sea ice concentration and are based on the January-March averaged loading vector of the warming mode.





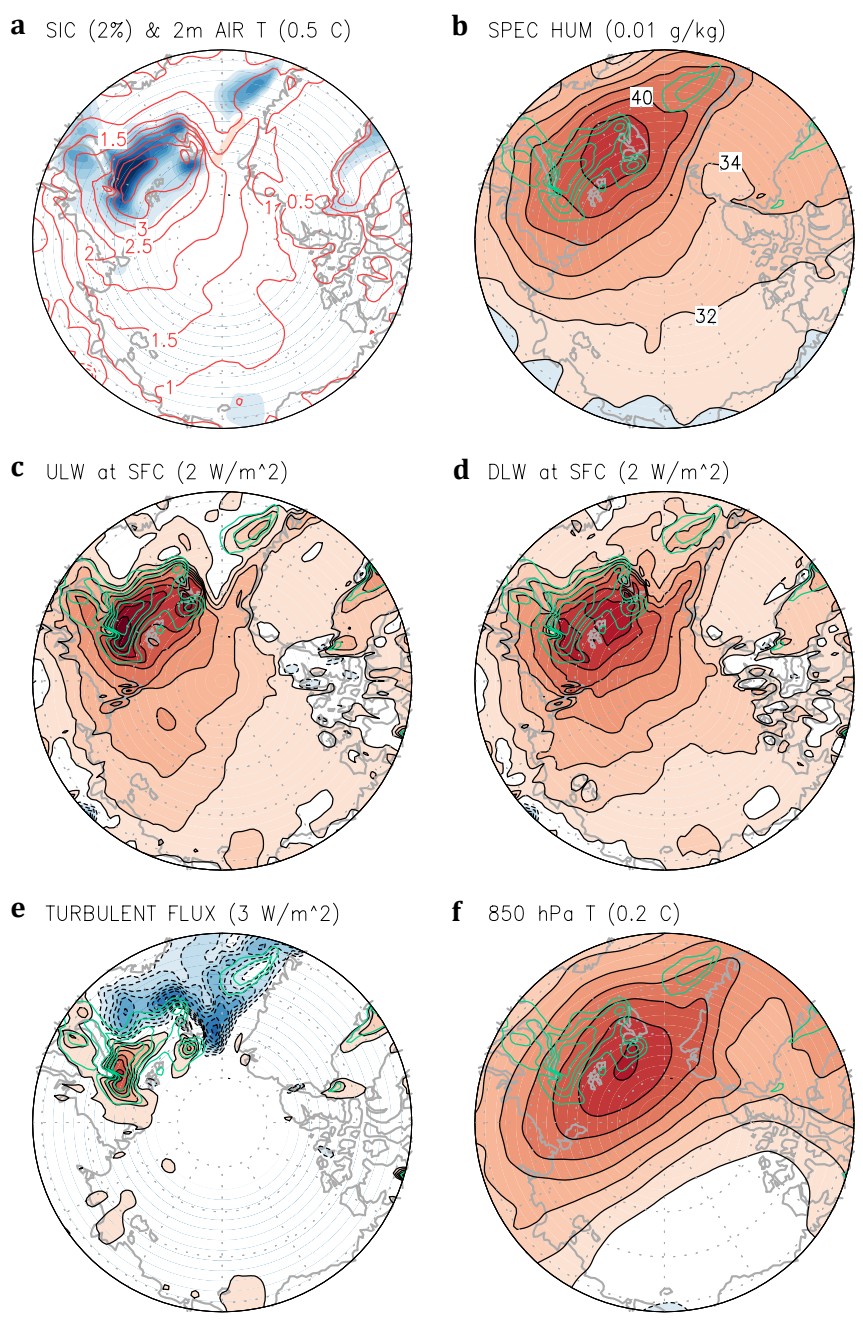

**Figure 10**. The regressed DJF patterns of (a) sea ice (shading) and 2 m air temperature (contour), (b) 900 hPa specific
humidity, (c) upward longwave radiation at surface, (d) downward longwave radiation at surface, (e) turbulent (sensible +
latent) heat flux, and (f) 850 hPa air temperature. The green contours in (b)-(f) represent sea ice concentration in (a).