# Peer review of "Mechanism of Seasonal Arctic Sea Ice Evolution and Arctic Amplification"

_The Cryosphere, 2016_

## Referee Comment (RC1) · Anonymous Referee #1 · 20 May 2016

As the Authors point out there is still considerable uncertainty as to the actual impact of a range of possible processes which could contribute to the so-called Arctic Amplification of the temperature trends. This paper potentially clarifies a number of these issues. The investigation reveals many interesting aspects of relevant feedbacks and the timescales of these. Valuable also is their insightful analysis of the different regional responses and why, e.g., the relative importance of these processes in the Barents and Kara Seas differs significantly from that in the Beaufort Sea.

The paper could potentially make a valuable contribution in TC. However I have identified places where the explanations need to be more accessible and some additional relevant literature be incorporated into the discussion.

PAGE 2, LINE 5 - 'Serreeze' should be 'Serreze'

PAGE 3, LINES 24-25 – The use of CSEOF is very interesting in this study, and it would be worth presenting a few more words of justification and explanation. I would suggest explicitly referring here to Kim et al. (2015) as it presents possibly a more approachable explanation of CSEOF and illustrates this with useful examples.

PAGE 4, LINE 7 – 'physically related' is perhaps a more accurate expression than 'physically consistent'

PAGE 5, LINE 15 - Do the authors mean 'well separated statistically' rather than 'well separated statistically'? Also, we need a quantitative backup for this statement. What test was used to establish independence (eg the North test)?

PAGE 6, LINES 18-20 - Another important aspect of this issue (involving another subtle positive feedback) which should also be mentioned here is that due to warming more Arctic precipitation is now falling as rain and less as snow, with strong implications for surface albedo. Refer here to analysis of Screen et al., 2012: Declining summer snowfall in the Arctic: Causes, impacts and feedbacks. Climate Dyn., 38, 2243-2256.

PAGE 9, LINES 7-12 - Also refer here to the recent new insights of Ärthun, M., and T. Eldevik, 2016: On anomalous ocean heat transport toward the Arctic and associated climate predictability. Journal of Climate, 29, 689-704, doi: 10.1175/jcli-d-15-0448.1.

PAGE 12, LINE 15 – Perhaps the authors have not fully appreciated the interpretation of the results in Vladimir Alexeev's paper. Central to their study is the presence of AA in an aquaplanet model without sea ice feedbacks. This key result pertaining to the origins of AA should be mentioned much earlier in the paper.

PAGE 12, LINES 19-21 – The authors are making an excellent point here in connection with the role of synoptic systems in the Arctic region and their relationship with trends. Very valuable here to also reference Simmonds et al. 2008: Arctic climate change as manifest in cyclone behavior. J. Clim., 21, 5777-5796.

Need for correction in References ... PAGE 13 – Year of publication of Alexeev et al.
paper is 2005, not 2015 PAGE 15 – Last reference. Please note correct spelling of author name Sorteberg, A., and Walsh, J. E.

---

## Referee Comment (RC2) · Anonymous Referee #2 · 30 May 2016

This study applies a novel technique (Cyclostationary empirical orthogonal function analysis) to ERA-interim reanalysis to examine physical processes behind Arctic sea ice reductions and Arctic amplification. While the study is unique and has the potential to yield insight on causal mechanisms, a number of issues should be addressed before publication.

1. The CSEOF technique requires a more broad-based description, including how the approach differs from standard EOF analysis, interpretation of the results in Table 1, how the spatial and temporal components of Figure 2 were derived, and the impact of mode 1 explaining only 15% of the total variability.

2. Throughout the manuscript, the term 'sea ice melting' is used to describe what are essentially negative winter (DJF) sea ice concentration anomalies in the Kara/Barents

sea with a maximum value of -10% (Figure 3). I'm not convinced this is the correct terminology to use. The negative ice concentration anomalies in this region are impacted in some combination by the timing of sea ice formation and temperature anomalies during the ice growth season (thermodynamics) and potential changes in ice motion and advection during the winter (dynamics). This may be semantics, but to refer to winter season negative ice concentration anomalies as 'ice melting' does not seem appropriate to me.

3. Related to the point above, all of the analysis is based on ERA-interim reanalysis, including sea ice concentration. This raises some questions: -how is ERA-interim sea ice concentration derived? Has it been validated? How does it compare to the more widely used passive microwave sea ice concentration data records? -how sensitive is this analysis to the choice of sea ice information? Do the results differ if sea ice information independent of the ERA-interim atmospheric fields is used?

4. What is the source of the sea surface temperature data?

5. I understand the general idea behind calculating the 'sea ice melting mechanism' (Figure 9) in terms of feedbacks associated with a given sea ice concentration change. A 1% sea ice concentration change, however, is not really physically relevant. This is well within the error of ice concentration datasets, and sea ice doesn't really change in this manner. This feedback totally discounts the role of ice dynamics – ice doesn't simply sit in one place and respond to temperature anomalies with an increase or decrease in ice concentration.

---

## Referee Comment (RC3) · Anonymous Referee #3 · 15 Jun 2016

The authors have examined the mechanisms by which declining sea ice in the Arctic is contributing to Arctic amplification of climate change. They focused on the differential changes in the Barents-Kara, Laptev and Chukchi seas and identified a unique pattern of change in the Barents and Kara seas associated with turbulent transport of heat from open water in the winter. The authors make a useful contribution to our understanding of the surface energy budget under conditions of reduced sea ice in the Arctic.

I have a few general recommendations and some specific edits recommended:

General comments

1. The authors often talk about sea ice "melt" when they are referring to the trend toward reduced sea ice concentration. In some cases, "melt" may be the appropriate term, but in most cases it would be better to refer to reduced sea ice concentration

and/or extent. 2. The authors discuss the increase in 850hPa temperatures over the Barents and Kara seas of less than 0.1K, which they indicate leads to a ∼1 W/m2 in downwelling longwave, which leads to two questions (see p. 8, l. 25-34). a. How can the authors be certain that the change in 850hPa temperature is due to the reduced sea ice concentration? b. I did some back of the envelope calculations, and it appears the magnitude of the downwelling longwave change is too large to be fully explained by the 850hPa temperature increase. Perhaps increased atmospheric moisture is playing a role in the increased downwelling longwave? The authors briefly discuss (p. 9, l. 20-23) and cite the recent Park et al. paper (p. 12, l. 21-23), but perhaps this issue should be further explored/discussed. 3. I appreciated that the authors added a schematic to explain the processes involved in the Barents and Kara seas (Fig. 9). Unfortunately, I was still left somewhat confused by the figure. For example, the "Increase T 0.07K" does not indicate at what level. The arrows leave me wondering if this is all happening concurrently or if there is a time associated with each process. I think this schematic is a good idea but could benefit from some additional thought.

Minor comments

1. p. 1, l. 10: "Arctic" misspelled, article missing 2. p. 1, l. 14: remove "to be" 3. p. 2, l. 4: "Serreze" misspelled 4. p. 2, l. 7: "in the earlier period" please be specific 5. p. 2, l. 29: "remains to be melted" is awkward 6. p. 2, l. 32: "While summer sea ice melting is clearly seen..." Does this mean decreased summer sea ice concentrations? P. 3, l. 2: "winter sea ice melting" Is sea ice really melting during the winter? Please see general comment #1 above. 7. p. 3, l. 12-13: "each term in the feedback" is repeated 8. p. 3, l. 22: add "and" before "2 m temperature" 9. p. 5, l. 4: "extract physically meaningful consistent evolutions from these variables" I was confused by this statement, perhaps because of the use of the word "evolutions" 10. p. 5, l. 9: "volatile" is not the word choice I would have expected 11. p. 6, l. 2: should be "increases" (agr) 12. p. 7, l. 21-22: "It is noted that" and "It is also worthy of remark that" are not necessary 13. p. 8, l. 2: "is maintaining sea ice stay melted" is confusing 14. p. 9, l. 21: "trapping" is not

a good description of this process 15. Figure 3 (and others): Some of the contours are difficult to follow, particularly on the JJA panel. If you chose not to label some of the contours, you may want to indicate the contour interval in the captions.

---

## Author Comment (AC1) · 28 Jun 2016

Interactive comment on "Mechanism of Seasonal Arctic Sea Ice Evolution and Arctic Amplification" by K.-Y. Kim et al.

Anonymous Referee #1

As the Authors point out there is still considerable uncertainty as to the actual impact of a range of possible processes which could contribute to the so-called Arctic Amplification of the temperature trends. This paper potentially clarifies a number of these issues. The investigation reveals many interesting aspects of relevant feedbacks and the timescales of these. Valuable also is their insightful analysis of the different regional responses and why, e.g., the relative importance of these processes in the Barents and Kara Seas differs significantly from that in the Beaufort Sea. The paper could potentially make a valuable contribution in TC. However I have identified places where the explanations need to be more accessible and some additional relevant literature be incorporated into the discussion.

Comment1(C1): PAGE 2, LINE 5 - 'Serreeze' should be 'Serreze'

Response1(R1): Corrected. [P2 L5]

C2: PAGE 3, LINES 24-25 – The use of CSEOF is very interesting in this study, and it would be worth presenting a few more words of justification and explanation. I would suggest explicitly referring here to Kim et al. (2015) as it presents possibly a more approachable explanation of CSEOF and illustrates this with useful examples.

R2: The reference was added in the revised manuscript. [P4 L8] We also revised the "method of analysis" section significantly so that the discussion therein is easier to understand. [P4 L7 – P6 L13]

C3: PAGE 4, LINE 7 – 'physically related' is perhaps a more accurate expression than 'physically consistent'

R3: The two loading vectors {B_n (r,t)} and {C_n (r,t)} are physically consistent in the context of the governing equation connecting the two variables. In other words, {B_n (r,t)} and {C_n (r,t)} satisfy the governing equation. Thus, the loading vectors are more than "physically related". Please let us keep the expression "physically consistent".

C4: PAGE 5, LINE 15 - Do the authors mean 'well separated statistically' rather than 'well separated dynamically'? Also, we need a quantitative backup for this statement. What test was used to establish independence (eg the North test)?

R4: The first CSEOF mode represents Arctic Amplification, whereas the second CSEOF mode represents Arctic Oscillation [Kim, K.-Y. and Son, S.-W.: Physical characteristics of Eurasian winter temperature variability, Environ. Res. Lett., 11, 044009, 2016.]. There is no such test as "North's rule of thumb" in CSEOF analysis. The Arctic warming mode is nearly independent with the second and third leading modes as can

be seen in Figure R1.

Figure R1; two time series are independent if correlation is zero for any lag. As can be seen in Figure R1, correlation is sufficiently low (in terms of R2 value, it is less than 0.1). In other words, we cannot explain the first PC time series reasonably by lagging either the second PC time series or the third PC time series. Thus, the Arctic warming mode is (nearly) independent of the next two CSEOF modes. We added the reference Kim and Son (2016). [P7 L17: . . . averaged AO index (Kim and Son, 2016).]

C5: PAGE 6, LINES 18-20 - Another important aspect of this issue (involving another subtle positive feedback) which should also be mentioned here is that due to warming more Arctic precipitation is now falling as rain and less as snow, with strong implications for surface albedo. Refer here to analysis of Screen et al., 2012: Declining summer snowfall in the Arctic: Causes, impacts and feedbacks. Climate Dyn., 38, 2243-2256.

R5: The reference was added in the text. [P8 L24, reference added]

C6: PAGE 9, LINES 7-12 - Also refer here to the recent new insights of Ärthun, M., and T. Eldevik, 2016: On anomalous ocean heat transport toward the Arctic and associated climate predictability. Journal of Climate, 29, 689-704, doi: 10.1175/jcli-d-15-0448.1.

R6: The reference was added in the text. [P11 L10, reference added]

C7: PAGE 12, LINE 15 – Perhaps the authors have not fully appreciated the interpretation of the results in Vladimir Alexeev's paper. Central to their study is the presence of AA in an aquaplanet model without sea ice feedbacks. This key result pertaining to the origins of AA should be mentioned much earlier in the paper.

R7: Correct. We made a mistake. We removed this discussion.

C8: PAGE 12, LINES 19-21 – The authors are making an excellent point here in connection with the role of synoptic systems in the Arctic region and their relationship with trends. Very valuable here to also reference Simmonds et al. 2008: Arctic climate change as manifest in cyclone behavior. J. Clim., 21, 5777-5796.

R8: Thank you for your suggestion. We added the reference. [P14 L19]

C9: Need for correction in References . . . PAGE 13 – Year of publication of Alexeev et al. paper is 2005, not 2015

R9: The reference was removed as mentioned in comment #7.

C10: PAGE 15 – Last reference. Please note correct spelling of author name Sorteberg, A., and Walsh, J. E.

R10: Author's name has been corrected. [reference]

* The combined response file including a marked-up manuscript is attached.

Please also note the supplement to this comment:
http://www.the-cryosphere-discuss.net/tc-2016-69/tc-2016-69-AC1-supplement.pdf

[Figure]

**Fig. 1.** Cross-correlation of the first PC time series against the second (red) and third (blue) PC time series for the lag range of (-24,24).

**Supplement:**

5

As the Authors point out there is still considerable uncertainty as to the actual impact of a range of possible processes which could contribute to the so-called Arctic Amplification of the temperature trends. This paper potentially clarifies a number of these issues. The investigation reveals many interesting aspects of relevant feedbacks and the timescales of these. Valuable also is their insightful analysis of the different regional responses and why, e.g., the relative importance of these processes in the Barents and Kara Seas differs significantly from that in the Beaufort Sea. The paper could potentially make a valuable contribution in TC. However I have identified places where the explanations need to be more accessible and some additional relevant literature be incorporated into the discussion.

Comment1(C1): PAGE 2, LINE 5 - 'Serreeze' should be 'Serreze'

15

10

Response1(R1): Corrected. [P2 L5]

C2: PAGE 3, LINES 24-25 - The use of CSEOF is very interesting in this study, and it would be worth presenting a few more words of justification and explanation. I would suggest explicitly referring here to Kim et al. (2015) as it presents
20 possibly a more approachable explanation of CSEOF and illustrates this with useful examples.

R2: The reference was added in the revised manuscript. [P4 L8] We also revised the "method of analysis" section significantly so that the discussion therein is easier to understand. [P4 L7 – P6 L13]

25

30

C3: PAGE 4, LINE 7 - 'physically related' is perhaps a more accurate expression than 'physically consistent'

R3: The two loading vectors  $\{B_n(r,t)\}$  and  $\{C_n(r,t)\}$  are physically consistent in the context of the governing equation connecting the two variables. In other words,  $\{B_n(r,t)\}$  and  $\{C_n(r,t)\}$  satisfy the governing equation. Thus, the loading vectors are more than "physically related". Please let us keep the expression "physically consistent".

C4: PAGE 5, LINE 15 - Do the authors mean 'well separated statistically' rather than 'well separated dynamically? Also, we need a quantitative backup for this statement. What test was used to establish independence (eg the North test)?

R4: The first CSEOF mode represents Arctic Amplification, whereas the second CSEOF mode represents Arctic Oscillation [Kim, K.-Y. and Son, S.-W.: Physical characteristics of Eurasian winter temperature variability, Environ. Res. Lett., 11, 044009, 2016.]. There is no such test as "North's rule of thumb" in CSEOF analysis. The Arctic warming mode is nearly independent with the second and third leading modes as can be seen in

Figure R1. Cross-correlation of the first PC time series against the second (red) and third (blue) PC time series for the lag range of (-24,24).

15

Figure R1; two time series are independent if correlation is zero for any lag. As can be seen in Figure R1, correlation is sufficiently low (in terms of  $R^2$  value, it is less than 0.1). In other words, we cannot explain the first PC time series reasonably by lagging either the second PC time series or the third PC time series. Thus, the Arctic warming mode is (nearly) independent of the next two CSEOF modes. We added the reference Kim and Son (2016). [P7 L17: ... averaged AO index (Kim and Son, 2016).]

C5: PAGE 6, LINES 18-20 - Another important aspect of this issue (involving another subtle positive feedback) which should also be mentioned here is that due to warming more Arctic precipitation is now falling as rain and less as snow, with strong implications for surface albedo. Refer here to analysis of Screen et al., 2012: Declining summer snowfall in the
 Arctic: Causes, impacts and feedbacks. Climate Dyn., 38, 2243-2256.

R5: The reference was added in the text. [P8 L24, reference added]

C6: PAGE 9, LINES 7-12 - Also refer here to the recent new insights of Ärthun, M., and T. Eldevik, 2016: On anomalous
ocean heat transport toward the Arctic and associated climate predictability. Journal of Climate, 29, 689-704, doi: 10.1175/jcli-d-15-0448.1.

R6: The reference was added in the text. [P11 L10, reference added]

C7: PAGE 12, LINE 15 – Perhaps the authors have not fully appreciated the interpretation of the results in Vladimir
5 Alexeev's paper. Central to their study is the presence of AA in an aquaplanet model without sea ice feedbacks. This key result pertaining to the origins of AA should be mentioned much earlier in the paper.

R7: Correct. We made a mistake. We removed this discussion.

10 C8: PAGE 12, LINES 19-21 – The authors are making an excellent point here in connection with the role of synoptic systems in the Arctic region and their relationship with trends. Very valuable here to also reference Simmonds et al. 2008: Arctic climate change as manifest in cyclone behavior. J. Clim., 21, 5777-5796.

R8: Thank you for your suggestion. We added the reference. [P14 L19]

15

C9: Need for correction in References . . . PAGE 13 - Year of publication of Alexeev et al. paper is 2005, not 2015

3

R9: The reference was removed as mentioned in comment #7.

20 C10: PAGE 15 - Last reference. Please note correct spelling of author name Sorteberg, A., and Walsh, J. E.

R10: Author's name has been corrected. [reference]

**Mechanism of Seasonal Arctic Sea Ice Evolution and Arctic Amplification**

Kwang-Yul Kim1\*, Benjamin D. Hamlington2, Hanna Na3, and Jinju Kim1

1School of Earth and Environmental Sciences, Seoul National University, Seoul 151-742, Republic of Korea

2Department of Ocean, Earth and Atmospheric Sciences, Old Dominion University, Norfork, Virginia 23529, United States of America

3Ocean Circulation and Climate Research Center, Korea Institute of Ocean Science and Technology, Ansan, 15627, Republic of Korea

Correspondence to: Kwang-Yul Kim (kwang56@snu.ac.kr)

5

- 10 Abstract. Sea ice loss is proposed as a primary reason for the Arctic amplification, although physical mechanism of the Arctic amplification and its connection with sea ice melting is still in debate. In the present study, monthly ERA-interim reanalysis data are analyzed via cyclostationary empirical orthogonal function analysis to understand the seasonal mechanism of sea ice loss in the Arctic Ocean and the Arctic amplification. While sea ice loss is widespread over much of the perimeter of the Arctic Ocean in summer, sea ice remains thin in winter only in the Barents-Kara Seas. Excessive
- 15 turbulent heat flux through the sea surface exposed to air due to sea ice reduction warms the atmospheric column. Warmer air increases the downward longwave radiation and subsequently surface air temperature, which facilitates sea surface remains to be free of ice. This positive feedback mechanism is not clearly observed in the Laptev, East Siberian, Chukchi, and Beaufort Seas, since sea ice refreezes in late fall (November) before excessive turbulent heat flux is available for warming the atmospheric column in winter. A detailed seasonal heat budget is presented in order to understand specific

4

20 differences between the Barents-Kara Seas and Laptev, East Siberian, Chukchi, and Beaufort Seas.

| Jinju Kim 6/22/2016 11:12 AM                                                                                                                                                      |
|-----------------------------------------------------------------------------------------------------------------------------------------------------------------------------------|
| Deleted: melting                                                                                                                                                                  |
| Jinju Kim 6/22/2016 11:13 AM                                                                                                                                                      |
| Deleted: Artic                                                                                                                                                                    |
| Jinju Kim 6/22/2016 11:13 AM                                                                                                                                                      |
| Deleted: melting                                                                                                                                                                  |
| Jinju Kim 6/22/2016 11:13 AM                                                                                                                                                      |
| Deleted: melting                                                                                                                                                                  |
| Jinju Kim 6/27/2016 2:44 PM                                                                                                                                                       |
| Deleted: to be                                                                                                                                                                    |
| Jinju Kim 6/22/2016 11:13 AM                                                                                                                                                      |
| Deleted: melting                                                                                                                                                                  |
| Kwang Yul Kim 6/27/2016 9:39 AM                                                                                                                                                   |
| Deleted: loss                                                                                                                                                                     |
| Jinju Kim 6/27/2016 4:00 PM                                                                                                                                                       |
| Deleted: ice                                                                                                                                                                      |
| Kwang Yul Kim 6/27/2016 9:39 AM                                                                                                                                                   |
| Deleted: A 1% reduction in sea ice concentration
in winter leads to $\sim$ 0.76 W m 2 increase in upward
heat flux, $\sim$ 0.07 K increase in 850 hPa air |

between  $A = 70^{-10}$  relation in Scarce concentration in winter leads to  $\sim 0.76 \text{ Wm}^2$  increase in upward heat flux,  $\sim 0.07 \text{ K}$  increase in 850 hPa air temperature,  $\sim 0.97 \text{ Wm}^2$  increase in downward longwave radiation, and  $\sim 0.26 \text{ K}$  increase in surface air temperature.

**1. Introduction**

[revised manuscript text omitted]

- mechanism of sea ice reduction and Arctic amplification is investigated from both a spatial and temporal standpoint, so that any delayed response can be explicitly considered. Quantification of each term in the feedback process is attempted in order to clarify their relative importance in the feedback. Further, the role of water vapor and cloud in the feedback process is assessed. Another key issue to be addressed is why and how sea ice Joss in winter develops in the Barents and Kara Seas but
- 15 not in the Laptev and Chukchi Seas. This issue is important in order to understand the key components of and reduce uncertainty in the feedback process. Also, it is pivotal to determine how fast the Arctic amplification progresses. The rate of acceleration of the Arctic amplification is estimated based on CSEOF analysis.

6

Jinju Kim 6/22/2016 11:17 AM Deleted: melting Jinju Kim 6/27/2016 3:08 PM Deleted: melting

Jinju Kim 6/22/2016 11:17 AM **Deleted:** melting

**Jinju Kim 6/22/2016 11:17 AM **Deleted:** melting**

Jinju Kim 6/22/2016 11:18 AM **Deleted:** the Jinju Kim 6/22/2016 11:18 AM **Deleted:** of each term Jinju Kim 6/22/2016 11:17 AM **Deleted:** melting

**2. Data and Method of Analysis**

The dataset used in the present study is the ERA-interim  $1.5^{\circ}\times1.5^{\circ}$  monthly reanalysis (Dee et al., 2011) from 1979-2014. Surface variables analyzed in the present study include sea surface temperature, sea ice concentration, latent and sensible heat fluxes, upward and downward longwave and shortwave radiations, and 2 m air temperature. Pressure-level variables analyzed include air temperature, geopotential, zonal wind, meridional wind, and specific humidity. Low-level and total cloud fractions are also analyzed.

The analysis tool employed in this study is the cyclostationary EOF (CSEOF) technique (Kim et al., 1996; Kim and North, 1997; Kim et al., 2015). In CSEOF analysis, data T(r, t) are decomposed in the form

$$T(r,t) = \sum_{n} B_n(r,t) T_n(t)$$

 $B_n(r,t) = B_n(r,t+d),$

(1)

10 where  $B_n(r, t)$  are mutually orthogonal CSEOF loading vectors (CSLV) and  $T_n(t)$  are mutually uncorrelated principal component (PC) time series of variable T(r, t). As in EOF analysis, a main motivation of CSEOF analysis is to decompose variability into uncorrelated and orthogonal components in order to understand major constituents of variability in T(r, t). Unlike EOF loading vector which is a spatial pattern, CSLV is a function of space and time describing temporal evolution pertaining to a physical process in T(r, t). Further, CSLV is periodic in time

5

(2)

where the periodicity d is called the nested period. This periodicity derives from the cyclostationarity assumption that the statistics of T(r, t) is periodic. For example, space-time covariance function of T(r, t) is defined by

$$C(r,t;r',t') = \langle T(r,t)T(r',t') \rangle = C(r,t+d;r',t'+d).$$
(3)

CSEOF loading vectors are derived as eigenvectors of periodic space-time covariance function by solving

$$\underline{\qquad \qquad } \mathcal{C}(r,t;r',t') \cdot \mathcal{B}_n(r',t') = \lambda_n \mathcal{
[revised manuscript text omitted]

Jinju Kim 6/27/2016 11:02 AM

... [2]

Jinju Kim 6/22/2016 11:28 AM Deleted: melting

Jinju Kim 6/22/2016 11:28 AM **Deleted:** melting

Kwang Yul Kim 6/27/2016 9:40 AM Deleted: 10 Jinju Kim 6/27/2016 4:14 PM Deleted: ice

Kwang Yul Kim 6/27/2016 9:40 AM Deleted: 10f Kwang Yul Kim 6/27/2016 9:40 AM Deleted: 10d

Jinju Kim 6/22/2016 11:28 AM Deleted: melting Kwang Yul Kim 6/27/2016 9:41 AM Deleted: 10 Jinju Kim 6/22/2016 11:28 AM Deleted: melting

In the Barents and Kara Seas, upward heat flux is increased due to the reduction in sea ice concentration in winter. This flux may be used to warm the lower troposphere, which, in turn, increases downward longwave radiation. As a result, surface air temperature may increase, which helps maintain the ice-free condition (see also Fig. 2). Such a mechanism persists throughout the winter, since sea ice does not refreeze, at least completely, until turbulent heat flux is sufficiently increased

- during cold winter. Specific humidity increases as atmospheric temperature increases; the anomalous patterns of the two are highly correlated. Thus, it appears that the increased specific humidity may have also contributed to the increase in downward longwave radiation. The anomalous pattern of cloud cover, however, is not significantly correlated with that of atmospheric temperature, suggesting that change in cloud cover has not significantly contributed to the Arctic amplification. The physical process of sea ice loss and increased air temperature appears to have been accelerating. According to a simple
- exponential fitting to the PC time series of the warming mode, the strength of this positive feedback process increases by ~8 % every year. At this rate, surface air temperature (850 hPa temperature) may increase by ~10 K (~3 K) over the Barents and Kara Seas with respect to the 1979 winter mean value as sea ice completely disappears (see also IPCC, 2013).

It should be pointed out that several different mechanisms have been invoked to explain Arctic amplification. For example, Hall (2004), Graversen and Wang (2009), and Graversen et al. (2014) showed based on model experiments that surface albedo feedback explains a large fraction of polar temperature amplification. Pithan and Mauritsen (2014) and Graversen et al. (2014) demonstrated that lapse-rate feedback also contributes to polar amplification using climate models. Finally, it should be pointed out that there are different mechanisms by which atmospheric moisture can be transported to the Barents and Kara Seas. For example, Sorteberg and Walsh (2008) demonstrated that moisture transport into the Arctic has increased due to increased seasonal cyclonic activity. Simmonds and Keay (2009) and Simmonds et al. (2008) showed that the trends and variability in September ice coverage is related to the mean cyclone characteristics. Park et al. (2015) showed that downward infrared radiation in the Arctic is driven by horizontal atmospheric water flux and warm air advection into the Arctic. Simmonds and Govekar (2014) also argued that sea ice reduction in the Arctic may be due to the advection of warm and humid air into the Arctic. In light of different views on sea ice loss and temperature amplification in the Arctic, the present study should be understood as a contrasting and complementary view on the mechanism of sea ice loss and

25 temperature amplification in the Arctic.

Data and code Availability: All the results of analysis and the programs used in the present paper are freely available by

contacting the corresponding author.

30 Acknowledgments: This research was supported by SNU-Yonsei Research Cooperation Program through Seoul National University in 2015.

| Kwang Yul Kim 6/27/2016 9:45 AM                                                                                                                                                                                                                                                                                                                                                                                                                                                                       |
|-------------------------------------------------------------------------------------------------------------------------------------------------------------------------------------------------------------------------------------------------------------------------------------------------------------------------------------------------------------------------------------------------------------------------------------------------------------------------------------------------------|
| Deleted: a 1 %                                                                                                                                                                                                                                                                                                                                                                                                                                                                                        |
| Kwang Yul Kim 6/27/2016 9:47 AM                                                                                                                                                                                                                                                                                                                                                                                                                                                                       |
| Deleted: leads to $\sim 0.76 \text{ W m}^{-2}$ increase in upward heat flux                                                                                                                                                                                                                                                                                                                                                                                                                    |
| Jinju Kim 6/27/2016 4:15 PM                                                                                                                                                                                                                                                                                                                                                                                                                                                                           |
| Deleted: excessive                                                                                                                                                                                                                                                                                                                                                                                                                                                                                    |
| Kwang Yul Kim 6/27/2016 9:48 AM                                                                                                                                                                                                                                                                                                                                                                                                                                                                       |
| Deleted: is                                                                                                                                                                                                                                                                                                                                                                                                                                                                                           |
| Kwang Yul Kim 6/27/2016 9:48 AM                                                                                                                                                                                                                                                                                                                                                                                                                                                                       |
| Deleted: ; 850 hPa air temperature increases by ~0.07 K. The raised air temperature                                                                                                                                                                                                                                                                                                                                                                                                            |
| Kunana a Mul Kina 0/07/0040 0.40 AM                                                                                                                                                                                                                                                                                                                                                                                                                                                                   |
| Kwang Yui Kim 6/27/2016 9:48 AM                                                                                                                                                                                                                                                                                                                                                                                                                                                                       |
| Deleted: by ~0.97 W m -2                                                                                                                                                                                                                                                                                                                                                                                                                                                            |
| Deleted:         by ~0.97 W m 2 Kwang Yul Kim 6/27/2016 9:50 AM                                                                                                                                                                                                                                                                                                                                                                                                                            |
| Deleted:         by~0.97 W m²           Kwang Yul Kim 6/27/2016 9:50 AM           Deleted:         s by~0.26 K                                                                                                                                                                                                                                                                                                                                                                                        |
| Deleted:         by ~0.97 W m 2 Kwang Yul Kim 6/27/2016 9:50 AM           Deleted:         s by ~0.26 K           Kwang Yul Kim 6/27/2016 9:41 AM                                                                                                                                                                                                                                                                                                                                          |
| Kwang Yui Kim 6/27/2016 9:48 AM           Deleted:         by ~0.97 W m²           Kwang Yui Kim 6/27/2016 9:50 AM           Deleted:         s by ~0.26 K           Kwang Yui Kim 6/27/2016 9:41 AM           Deleted:         10                                                                                                                                                                                                                                                                    |
| Kwang Yui Kim 6/27/2016 9:48 AM           Deleted:         by ~0.97 W m²           Kwang Yui Kim 6/27/2016 9:50 AM           Deleted:         s by ~0.26 K           Kwang Yui Kim 6/27/2016 9:41 AM           Deleted:         10           Jinju Kim 6/22/2016 11:28 AM                                                                                                                                                                                                                             |
| Kwang Yui Kim 6/27/2016 9:48 AM           Deleted:         by ~0.97 W m²           Kwang Yui Kim 6/27/2016 9:50 AM           Deleted:         s by ~0.26 K           Kwang Yui Kim 6/27/2016 9:41 AM           Deleted:         10           Jinju Kim 6/22/2016 11:28 AM           Deleted:         melting                                                                                                                                                                                          |
| Kwang Yui Kim 6/27/2016 9:48 AM           Deleted:         by ~0.97 W m²           Kwang Yui Kim 6/27/2016 9:50 AM           Deleted:         s by ~0.26 K           Kwang Yui Kim 6/27/2016 9:41 AM           Deleted:         10           Jinju Kim 6/22/2016 11:28 AM           Deleted:         melting           Jinju Kim 6/22/2016 11:28 AM                                                                                                                                                   |
| Kwang Yui Kim 6/27/2016 9:48 AM           Deleted:         by ~0.97 W m²           Kwang Yui Kim 6/27/2016 9:50 AM           Deleted:         s by ~0.26 K           Kwang Yui Kim 6/27/2016 9:41 AM           Deleted:         10           Jinju Kim 6/22/2016 11:28 AM           Deleted:         melting           Jinju Kim 6/22/2016 11:28 AM           Deleted:         melting                                                                                                                |
| Kwang Yui Kim 6/27/2016 9:48 AM           Deleted:         by ~0.97 W m²           Kwang Yui Kim 6/27/2016 9:50 AM           Deleted:         s by ~0.26 K           Kwang Yui Kim 6/27/2016 9:41 AM           Deleted:         10           Jinju Kim 6/22/2016 11:28 AM           Deleted:         melting           Jinju Kim 6/22/2016 11:28 AM           Deleted:         melting           Jinju Kim 6/22/2016 11:28 AM           Deleted:         melts           Jinju Kim 6/22/2016 11:29 AM |

Jinju Kim 6/27/2016 10:58 AM Deleted:

Jinju Kim 6/27/2016 6:48 PM Deleted: e

Jinju Kim 6/27/2016 6:45 PM Deleted: 2010 Jinju Kim 6/27/2016 6:44 PM Deleted: Y

Jinju Kim 6/27/2016 6:40 PM Deleted: ts

Jinju Kim 6/28/2016 9:13 AM Deleted: -

[revised manuscript text omitted]

---

## Author Comment (AC2) · 28 Jun 2016

Interactive comment on "Mechanism of Seasonal Arctic Sea Ice Evolution and Arctic Amplification" by K.-Y. Kim et al.

Anonymous Referee #2

This study applies a novel technique (Cyclostationary empirical orthogonal function analysis) to ERA-interim reanalysis to examine physical processes behind Arctic sea ice reductions and Arctic amplification. While the study is unique and has the potential to yield insight on causal mechanisms, a number of issues should be addressed before publication.

Comment1(C1): The CSEOF technique requires a more broad-based description, including how the approach differs from standard EOF analysis, interpretation of the results in Table 1, how the spatial and temporal components of Figure 2 were derived, and the impact of mode 1 explaining only 15% of the total variability.

Response1(R1): We modified the "method of analysis" section significantly in order to address what the reviewer requested including the definition and interpretation of R2 value. This section should be much clearer now. [P4 L7- P6 L13]

The spatial patterns and temporal components of Figure 2 (in manuscript) are the result of CSEOF analysis. They are $B_1$ (r,t) and $T_1$ (t) of 2m air temperature.

The leading mode explains ∼15% of the total variability of 2 m air temperature. In other words, 85% of the variance of surface air temperature variability derives from other mechanisms. This is something that we cannot control; it is the nature of our climate system. Since temperature variability associated with Arctic amplification explains only ∼15% of the total variability, it is all the more important to separate it from other mechanisms of variability. Otherwise, temperature variability associated with Arctic amplification will be obscured seriously by variability from other sources, and an accurate physical and statistical inference of Arctic Amplification would be difficult. We also want to say that this mode is well separated from the second CSEOF mode (Arctic oscillation; Kim, K.-Y. and Son, S.-W.: Physical characteristics of Eurasian winter temperature variability, Environ. Res. Lett., 11, 044009, 2016.).

C2: Throughout the manuscript, the term 'sea ice melting' is used to describe what are essentially negative winter (DJF) sea ice concentration anomalies in the Kara/Barents sea with a maximum value of -10% (Figure 3). I'm not convinced this is the correct terminology to use. The negative ice concentration anomalies in this region are impacted in some combination by the timing of sea ice formation and temperature anomalies during the ice growth season (thermodynamics) and potential changes in ice motion and advection during the winter (dynamics). This may be semantics, but to refer to winter season negative ice concentration anomalies as 'ice melting' does not seem

appropriate to me.

R2: We used the terminology 'sea ice melting' in order to address sea ice loss from the perspective of Arctic amplification. As you mentioned, however, melting is not the only means of sea ice reduction. Therefore, we changed 'sea ice melting' to 'sea ice loss' or 'sea ice reduction' except when we really meant 'melting'. [correction scatted in the manuscript]

C3: Related to the point above, all of the analysis is based on ERA-interim reanalysis, including sea ice concentration. This raises some questions: -how is ERA-interim sea ice concentration derived? Has it been validated? How does it compare to the more widely used passive microwave sea ice concentration data records? -how sensitive is this analysis to the choice of sea ice information? Do the results differ if sea ice information independent of the ERA-interim atmospheric fields is used?

R3: Sea ice concentration (SIC) in the ERA-Interim data is an estimate from other operational products such as operational NCEP product and sea surface temperature and sea ice analysis (OSTIA). These products are based on passive microwave satellite measurements (Donlon et al. : The Operational Sea Surface Temperature and Sea Ice Analysis (OSTIA) system, Remote Sens. Environ., 116, 140-158, 2012). In order to confirm that the results in the present study are not sensitive to the sea ice concentration dataset, CSEOF analysis was conducted on a different sea ice concentration dataset acquired from National Snow & Ice Data Center (NSIDC at nsidc.org). The specific dataset used is "sea ice concentration from Nimbus-7 SSMR and DMSP SSM/I-SSMIS Passive Microwave data", which is generated from brightness temperature data. The data are provided in the polar stereographic projection at a grid cell size of 25 x 25 km. We used the data in 1990-2014, since there are a large number of missing points in earlier data.

Figure R1 is the regressed patterns of sea ice concentration for the first (Arctic Amplification) CSEOF mode derived from the NSIDC dataset in comparison with that derived

from the ERA-Interim data. As can be seen in the figure, there is no serious discrepancy between the two datasets. Both the spatial pattern and magnitude of variation in sea ice concentration in conjunction with the Arctic Amplification mode is quite similar between the two. The R2 value of regression is 0.97 for the first CSEOF mode, suggesting that the amplitude of variation of the sea ice patterns in Figure R1 is physically consistent with the Arctic warming patterns in Figure 2 in the manuscript. [no corrective action]

C4: What is the source of the sea surface temperature data?

R4: Sea surface temperature of ERA-Interim reanalysis data has been used in the present study. As a comparison between Figures R3 and R4 shows, the SSTA patterns derived from the ERSST data are not much different from those of ERA-Interim reanalysis data in terms of the key features. We used the ERA-Interim SST in order to maintain physical consistency among the variables analyzed in the present study. [no corrective action]

C5: I understand the general idea behind calculating the 'sea ice melting mechanism' (Figure 9) in terms of feedbacks associated with a given sea ice concentration change. A 1% sea ice concentration change, however, is not really physically relevant. This is well within the error of ice concentration datasets, and sea ice doesn't really change in this manner. This feedback totally discounts the role of ice dynamics – ice doesn't simply sit in one place and respond to temperature anomalies with an increase or decrease in ice concentration.

R5: As the reviewer pointed out, sea ice loss is not entirely due to melting and ice dynamics certainly was not accounted for in this discussion. What we are referring to is an average picture of physical change due to sea ice loss. As explained in the method section, regression in CSEOF space allows us to write data in the form:

Data(r,t)=$\sum\_n\{B\_n(r,t), C\_reg\_n(r,t), D\_reg\_n(r,t), E\_reg\_n(r,t),\}T\_n(t),$

where the terms in curly braces for each n represents a physical process as reflected in different variables (say, temperature, sea ice concentration, 850 hPa air temperature, upward longwave radiation, etc.). The terms in curly braces are physically consistent with each other. For example, Figure R5 below shows the daily evolution associated with the Arctic Amplification mode. CSEOF analysis was conducted on the daily ERA-Interim data during winter (Dec. 1-Feb. 28), and the first CSEOF mode represents Arctic Amplification as in the present analysis. Shown in Figure R5 are the terms in curly braces for five different variables averaged over the Barents-Kara Seas [21°-79.5°E × 75°-79.5°N]. As can be seen in the figure, loss of sea ice is reflected in the positive values of anomalous 2 m air temperature, 850 hPa temperature, upward longwave radiation, and downward longwave radiation. Daily variations of atmospheric variables are highly correlated with each other, suggesting that they have a common cause (see Figure R6). Specifically, the impact of synoptic disturbance is conspicuous with significant fluctuations on synoptic time scales.

Further, CSEOF analysis indicates that these variations are amplifying in time as reflected in the PC time series in Figure R5(d). The mechanism described in Figure 9 is the winter average picture of the mechnism shown in Figure R5. We can average the CSLVs in Figure R5 during winter to estimate the relative magnitude of change in heat flux or atmospheric variables as sea ice loss continues. Whatever the cause of sea ice reduction is, a 1% loss of sea ice results in the changes in other variables as described in Figure 9. Further, the lagged correlation analysis among these variables indicates that turbulent heat flux preceeds 850 hPa warming, which, in turn, is followed by increased downward longwave radiation (see Figure R6). Ultimately, surface air temperature increases due to increased downward longwave radiation.

On the other hand, the mechanism addressed above cannot be demonstrated in CSEOF analysis of monthly data. The cause-and-effect relationship among the variables in Figure R5 can only be appreciated when we analyze 3-hourly data. Therefore, we remove the entire discussion associated with Figure 9. Hopefully, we will address

this mechanism in a new paper where 3-hourly data is employed for CSEOF analysis. [Removed the text associated with Figure 9.]

* Figure Captions

Figure R1. The Arctic Amplification mode of NSIDC sea ice concentration

Figure R2. The Arctic Amplification mode of ERA-Interim sea ice concentration

Figure R3. The regressed seasonal patterns of ERA-Interim sea surface temperature (shading; 0.05 K) and the reduction of sea ice concentration (contours; 2 %) in the Arctic region (64.5°-90° N).

Figure R4. The regressed seasonal patterns of ERSST (shading; 0.05 K) and the reduction of sea ice concentration (contours; 2 %) in the Arctic region (64.5°-90° N).

Figure R5. Daily patterns of variability over the region of sea ice loss (21°-79.5°E × 75°-79.5°N): (a) sea ice concentration, (b) 2 m air temperature (red), 850 hPa air temperature × 2 (black), and upward longwave radiation (blue), and (c) same as (b) except for the regressed downward longwave radiation (blue). Correlation of upward and downward longwave radiations with 2 m air temperature is respectively 0.90 and 0.95, whereas with 850 hPa air temperature is 0.60 and 0.86. (d) Corresponding PC time series.

Figure R6. Correlation of upward (solid lines) and downward (dotted lines) longwave radiations with 2 m air temperature (blue), 850 hPa temperature (red), and sea ice concentration (black). Longwave radiation lags the other variable for a positive lag. Lagged correlation between 2 m air temperature and 850 hPa air temperature (black dashed line); 2 m air temperature leads 850 hPa temperature for a positive lag.

** The combined response file including a marked-up manuscript is attached.

Please also note the supplement to this comment:
http://www.the-cryosphere-discuss.net/tc-2016-69/tc-2016-69-AC2-supplement.pdf

[Figure]

**Fig. 1.** The Arctic Amplification mode of NSIDC sea ice concentration

**Fig. 2.** The Arctic Amplification mode of ERA-Interim sea ice concentration

[Figure]

**Fig. 3.** The regressed seasonal patterns of ERA-Interim sea surface temperature (shading; 0.05 K) and the reduction of sea ice concentration (contours; 2 %) in the Arctic region (64.5°-90° N).

[Figure]

**Fig. 4.** The regressed seasonal patterns of ERSST (shading; 0.05 K) and the reduction of sea ice concentration (contours; 2 %) in the Arctic region (64.5°-90° N).

[Figure]

**Fig. 5.** Daily patterns of variability over the region of sea ice loss (21°-79.5°E × 75°-79.5°N):
(a) sea ice concentration, (b) 2 m air temperature (red), 850 hPa air temperature × 2 (black),
and upward longw

[Figure]

**Fig. 6.** Correlation of upward (solid lines) and downward (dotted lines) longwave radiations with 2 m air temperature (blue), 850 hPa temperature (red), and sea ice concentration (black). Longwave radiation l

**Supplement:**

This study applies a novel technique (Cyclostationary empirical orthogonal function analysis) to ERA-interim reanalysis to examine physical processes behind Arctic sea ice reductions and Arctic amplification. While the study is unique and has the potential to yield insight on causal mechanisms, a number of issues should be addressed before publication.

10  Comment1(C1): The CSEOF technique requires a more broad-based description, including how the approach differs from standard EOF analysis, interpretation of the results in Table 1, how the spatial and temporal components of Figure 2 were derived, and the impact of mode 1 explaining only 15% of the total variability.

Response1(R1): We modified the "method of analysis" section significantly in order to address what the reviewer requested
15  including the definition and interpretation of $R^2$ value. This section should be much clearer now. [P4 L7- P6 L13]

The spatial patterns and temporal components of Figure 2 are the result of CSEOF analysis. They are $B_1(r, t)$ and $T_1(t)$ of 2m air temperature.

20  The leading mode explains ~15% of the total variability of 2 m air temperature. In other words, 85% of the variance of surface air temperature variability derives from other mechanisms. This is something that we cannot control; it is the nature of our climate system. Since temperature variability associated with Arctic amplification explains only ~15% of the total variability, it is all the more important to separate it from other mechanisms of variability. Otherwise, temperature variability associated with Arctic amplification will be obscured seriously by variability from other sources, and an accurate
25  physical and statistical inference of Arctic Amplification would be difficult. We also want to say that this mode is well separated from the second CSEOF mode (Arctic oscillation; Kim, K.-Y. and Son, S.-W.: Physical characteristics of Eurasian winter temperature variability, Environ. Res. Lett., 11, 044009, 2016.).

C2: Throughout the manuscript, the term 'sea ice melting' is used to describe what are essentially negative winter (DJF) sea
30  ice concentration anomalies in the Kara/Barents sea with a maximum value of -10% (Figure 3). I'm not convinced this is the correct terminology to use. The negative ice concentration anomalies in this region are impacted in some combination by the timing of sea ice formation and temperature anomalies during the ice growth season (thermodynamics) and potential changes in ice motion and advection during the winter (dynamics). This may be semantics, but to refer to winter season negative ice concentration anomalies as 'ice melting' does not seem appropriate to me.

R2: We used the terminology 'sea ice melting' in order to address sea ice loss from the perspective of Arctic amplification. As you mentioned, however, melting is not the only means of sea ice reduction. Therefore, we changed 'sea ice melting' to 'sea ice loss' or 'sea ice reduction' except when we really meant 'melting'. [correction scatted in the manuscript]

C3: Related to the point above, all of the analysis is based on ERA-interim reanalysis, including sea ice concentration. This raises some questions: -how is ERA-interim sea ice concentration derived? Has it been validated? How does it compare to the more widely used passive microwave sea ice concentration data records? -how sensitive is this analysis to the choice of sea ice information? Do the results differ if sea ice information independent of the ERA-interim atmospheric fields is used?

R3: Sea ice concentration (SIC) in the ERA-Interim data is an estimate from other operational products such as operational NCEP product and sea surface temperature and sea ice analysis (OSTIA). These products are based on passive microwave satellite measuremnents (Donlon et al.: The Operational Sea Surface Temperature and Sea Ice Analysis (OSTIA) system, Remote Sens. Environ., 116, 140-158, 2012). In order to confirm that the results in the present study are not sensitive to the

15 sea ice concentration dataset, CSEOF analysis was conducted on a different sea ice concentration dataset acquired from National Snow & Ice Data Center (NSIDC at nsidc.org). The specific dataset used is "sea ice concentration from Nimbus-7 SSMR and DMSP SSM/I-SSMIS Passive Microwave data", which is generated from brightness temperature data. The data are provided in the polar stereographic projection at a grid cell size of 25 x 25 km. We used the data in 1990-2014, since there are a large number of missing points in earlier data.

Figure R1 is the regressed patterns of sea ice concentration for the first (Arctic Amplification) CSEOF mode derived from the NSIDC dataset in comparison with that derived from the ERA-Interim data. As can be seen in the figure, there is no serious discrepancy between the two datasets. Both the spatial pattern and magnitude of variation in sea ice concentration in conjunction with the Arctic Amplification mode is quite similar between the two. The $R^2$ value of regression is 0.97 for the

25 first CSEOF mode, suggesting that the amplitude of variation of the sea ice patterns in Figure R1 is physically consistent with the Arctic warming patterns in Figure 2 in the manuscript. [no corrective action]

[Figure]

**Figure R1**. The Arctic Amplification mode of NSIDC sea ice concentration

[Figure]

**Figure R2**. The Arctic Amplification mode of ERA-Interim sea ice concentration

C4: What is the source of the sea surface temperature data?

5    R4: Sea surface temperature of ERA-Interim reanalysis data has been used in the present study. As a comparison between Figures R3 and R4 shows, the SSTA patterns derived from the ERSST data are not much different from those of ERA-Interim reanalysis data in terms of the key features. We used the ERA-Interim SST in order to maintain physical consistency among the variables analyzed in the present study. [no corrective action]

[Figure]

**Figure R3**. The regressed seasonal patterns of ERA-Interim sea surface temperature (shading; 0.05 K) and the reduction of sea ice concentration (contours; 2 %) in the Arctic region (64.5°-90° N).

[Figure]

SST

**Figure R4**.   The regressed seasonal patterns of ERSST (shading; 0.05 K) and the reduction of sea ice concentration (contours; 2 %) in the Arctic region (64.5°-90° N).

C5: I understand the general idea behind calculating the 'sea ice melting mechanism' (Figure 9) in terms of feedbacks associated with a given sea ice concentration change. A 1% sea ice concentration change, however, is not really physically relevant. This is well within the error of ice concentration datasets, and sea ice doesn't really change in this manner. This feedback totally discounts the role of ice dynamics – ice doesn't simply sit in one place and respond to temperature anomalies with an increase or decrease in ice concentration.

R5: As the reviewer pointed out, sea ice loss is not entirely due to melting and ice dynamics certainly was not accounted for in this discussion. What we are referring to is an average picture of physical change due to sea ice loss. As explained in the method section, regression in CSEOF space allows us to write data in the form:

$$Data(r,t) = \sum_n \left\{ B_n(r,t), C_n^{(r)}(r,t), D_n^{(r)}(r,t), E_n^{(r)}(r,t), \cdots \right\} T_n(t),$$

5    where the terms in curly braces for each $n$ represents a physical process as reflected in different variables (say, temperature, sea ice concentration, 850 hPa air temperature, upward longwave radiation, etc.). The terms in curly braces are physically consistent with each other. For example, Figure R5 below shows the daily evolution associated with the Arctic Amplification mode. CSEOF analysis was conducted on the daily ERA-Interim data during winter (Dec. 1-Feb. 28), and the first CSEOF mode represents Arctic Amplification as in the present analysis. Shown in Figure R5 are the terms in curly

10   braces for five different variables averaged over the Barents-Kara Seas [21°-79.5°E × 75°-79.5°N]. As can be seen in the figure, loss of sea ice is reflected in the positive values of anomalous 2 m air temperature, 850 hPa temperature, upward longwave radiation, and downward longwave radiation. Daily variations of atmospheric variables are highly correlated with each other, suggesting that they have a common cause (see Figure R6). Specifically, the impact of synoptic disturbance is conspicuous with significant fluctuations on synoptic time scales.

Further, CSEOF analysis indicates that these variations are amplifying in time as reflected in the PC time series in Figure R5(d). The mechanism described in Figure 9 is the winter average picture of the mechnism shown in Figure R5. We can average the CSLVs in Figure R5 during winter to estimate the relative magnitude of change in heat flux or atmospheric variables as sea ice loss continues. Whatever the cause of sea ice reduction is, a 1% loss of sea ice results in the changes in

20   other variables as described in Figure 9. Further, the lagged correlation analysis among these variables indicates that turbulent heat flux preceeds 850 hPa warming, which, in turn, is followed by increased downward longwave radiation (see Figure R6). Ultimately, surface air temperature increases due to increased downward longwave radiation.

On the other hand, the mechanism addressed above cannot be demonstrated in CSEOF analysis of monthly data. The cause-

25   and-effect relationship among the variables in Figure R5 can only be appreciated when we analyze 3-hourly data. Therefore, we remove the entire discussion associated with Figure 9. Hopefully, we will address this mechanism in a new paper where 3-hourly data is employed for CSEOF analysis. [Removed the text associated with Figure 9.]

[Figure]

**Figure R5**. Daily patterns of variability over the region of sea ice loss (21°-79.5°E × 75°-79.5°N): (a) sea ice concentration, (b) 2 m air temperature (red), 850 hPa air temperature × 2 (black), and upward longwave radiation (blue), and (c) same as (b) except for the regressed downward longwave radiation (blue). Correlation of upward and downward longwave radiations with 2 m air temperature is respectively 0.90 and 0.95, whereas with 850 hPa air temperature is 0.60 and 0.86. (d) Corresponding PC time series.

[Figure]

**Figure R6**. Correlation of upward (solid lines) and downward (dotted lines) longwave radiations with 2 m air temperature (blue), 850 hPa temperature (red), and sea ice concentration (black). Longwave radiation lags the other variable for a positive lag. Lagged correlation between 2 m air temperature and 850 hPa air temperature (black dashed line); 2 m air temperature leads 850 hPa temperature for a positive lag.

**Mechanism of Seasonal Arctic Sea Ice Evolution and Arctic Amplification**

Kwang-Yul Kim[1*], Benjamin D. Hamlington[2], Hanna Na[3], and Jinju Kim[1]

[revised manuscript text omitted]

Kwang Yul Kim 6/27/2016 9:42 AM

---

## Author Comment (AC3) · 28 Jun 2016

Interactive comment on "Mechanism of Seasonal Arctic Sea Ice Evolution and Arctic Amplification" by K.-Y. Kim et al.

Anonymous Referee #3

The authors have examined the mechanisms by which declining sea ice in the Arctic is contributing to Arctic amplification of climate change. They focused on the differential changes in the Barents-Kara, Laptev and Chukchi seas and identified a unique pattern of change in the Barents and Kara seas associated with turbulent transport of heat from open water in the winter. The authors make a useful contribution to our understanding of the surface energy budget under conditions of reduced sea ice in the Arctic. I have a few general recommendations and some specific edits recommended:

General comments

Comment1(C1): The authors often talk about sea ice "melt" when they are referring to the trend toward reduced sea ice concentration. In some cases, "melt" may be the appropriate term, but in most cases it would be better to refer to reduced sea ice concentration and/or extent.

Response1(R1): Thank you for pointing this. We replaced "sea ice melting" by "sea ice loss" or "sea ice reduction" except when we really meant "melting". [Corrections are scattered throughout the manuscript.]

C2: The authors discuss the increase in 850hPa temperatures over the Barents and Kara seas of less than 0.1K, which they indicate leads to a âĹij1 W/m2 in downwelling longwave, which leads to two questions (see p. 8, l. 25-34). a. How can the authors be certain that the change in 850hPa temperature is due to the reduced sea ice concentration? b. I did some back of the envelope calculations, and it appears the magnitude of the downwelling longwave change is too large to be fully explained by the 850hPa temperature increase. Perhaps increased atmospheric moisture is playing a role in the increased downwelling longwave? The authors briefly discuss (p. 9, l. 20- 23) and cite the recent Park et al. paper (p. 12, l. 21-23), but perhaps this issue should be further explored/discussed.

R2a: As can be seen in Figure 10 (new Figure 9), the anomaly pattern of sea ice loss and those of turbulent heat flux, 2 m air temperature, upward longwave radiation, down-ward longwave radiation, and 850 hPa air temperature have common centers of action. These patterns share the same PC time series (Figure 2b). It seems reasonable to as-sume that these patterns share an identical source of variability. Figure R1 shows the loading vectors of the Arctic warming mode derived from the daily ERA-Interim data in winter (Dec. 1-Feb. 28; 90 days); the loading vectors are averaged over the Barents-Kara Seas. As can be seen in the figure, sea ice reduction is clearly seen throughout the winter (Figure R1a). The loading vector of turbulent heat flux has a positive mean

and is generally positive throughout the winter because of the sea ice reduction in the area (Figure R1b; red curve)). The 850 hPa air temperature anomaly is also positive with a mean value of $\sim$1.26 K (Figure R1b; black curve). Finally, the loading vector of specific humidity has a mean value of $\sim$0.15 g kg-1, and is highly correlated with the 850 hPa air temperature; correlation is 0.91 (Figure R1b; blue curve). The specific humidity is moderately correlated with moisture convergence (corr = 0.54). Turbulent heat flux is negatively correlated with 850 hPa air temperature and specific humidity. This negative correlation seems to indicate that turbulent heat flux decreases as air temperature increases and specific humidity increases and reflects the bulk formulas for latent and sensible heat flux. It should be noted that all the variables have significant positive means. This positive mean is due to sea ice reduction throughout the winter. On top of the increased turbulent heat flux through the open surface of the ocean, the release of the turbulent heat flux is affected by the atmospheric condition (such as air temperature and humidity) as well as horizontal moisture transport.

Figure R2 shows the winter average pattern of moisture transport and convergence for the Arctic warming mode. This pattern is similar to Figure 5 in Park et al. (2015). There is a sign of moisture convergence in the Barents-Kara Seas. On the other hand, the daily time series over the Barents-Kara Seas (Figure R1c) indicates that the sign of moisture convergence fluctuates around zero. Thus, it is difficult to explain the non-zero mean of specific humidity in Figure R1b in terms of the moisture transport and convergence.

Thus, we think that the loss of sea ice leads to increased turbulent heat flux, which not only warms the atmospheric column but also increases specific humidity. Saturation specific humidity also increases as the atmospheric column warms up. The increased air temperature and specific humidity both contribute to the increased downward long-wave radiation. This discussion is difficult to include in the revision, since it requires new analysis based on daily ERA-Interim reanalysis data. [no corrective action]

R2b: According to the first law of thermodynamics, we have

$$c_p \, \partial T/\partial t = - \text{Del} \cdot F , \quad (1)$$

where  is the density of air, $c_p$ is specific heat at constant pressure, T is temperature, and F âČŮ is heat flux. In a one-dimensional column model, (1) can be rewritten as

$$c_p \, \partial T/\partial t = -\partial/\partial z \, (F\_up - F\_down ) = -\partial/\partial z(F\_net) . \quad (2)$$

By integrating (2) with respect z from level $z_1$ to $z_2$, we have

$$c_p \, \partial/\partial t \, (\text{Integral}\_(z\_1)-(z\_2) \, T \, dz) = -F\_net \, (z\_2 ) + F\_net \, (z\_1) . \quad (3)$$

Let $z_1 = \varepsilon$ is the level at which radiative transfer is zero (say, slightly below the surface). Then, we can show that

$$c_p \, \partial/\partial t \, (\text{Integral}\_\varepsilon - (z\_2) \, T \, dz) = -F\_net \, (z\_2 ) = F\_down \, (z\_2 ) - F\_up \, (z\_2) . \quad (4)$$

If $z_2$ represents a vertical level slightly above the surface, the temperature near the surface is determined by downward and upward flux at level $z=z_2$. According to Figure 9, downward flux is larger than upward flux at the surface. Henceforth, surface temperature should increase. Likewise, we can let be the 850 hPa level and determine net heat flux for a temperature change integrated from 850 hPa to surface. Nonetheless, we cannot calculate upward flux and downward flux separately; we can only calculate net flux. Therefore, we cannot show what the downward flux from the 850 hPa level. In short, it is not the downward flux but the net flux that is related to the temperature change at the 850 hPa level (0.07 K). Of course, we do not have flux information at all vertical levels and we cannot verify theoretically that the downward flux 0.97 W m-2 is due to temperature change. [We eliminated Figure 9 together with the text associated with it.]

3. I appreciated that the authors added a schematic to explain the processes involved in the Barents and Kara seas (Fig. 9). Unfortunately, I was still left somewhat confused by the figure. For example, the "Increase T 0.07K" does not indicate at what level. The arrows leave me wondering if this is all happening concurrently or if there is a time associated with each process. I think this schematic is a good idea but could benefit

from some additional thought.

As explained in the method section, regression in CSEOF space allows us to write data in the form:

Data(r,t)=$\sum\_n\{B\_n(r,t), C\_reg\_n(r,t), D\_reg\_n(r,t), E\_reg\_n(r,t),\}T\_n(t)$,

where the terms in curly braces for each n represents a physical process as reflected in different variables (say, temperature, sea ice concentration, 850 hPa air temperature, upward longwave radiation, etc.). The terms in curly braces are physically consistent with each other. For example, Figure R3 below shows the daily evolution associated with the Arctic Amplification mode. CSEOF analysis was conducted on the daily ERA-Interim data during winter (Dec. 1-Feb. 28), and the first CSEOF mode represents Arctic Amplification as in the present analysis. Shown in Figure R3 are the terms in curly braces for five different variables averaged over the Barents-Kara Seas [21°-79.5°E × 75°-79.5°N]. As can be seen in the figure, loss of sea ice is reflected in the positive values of anomalous 2 m air temperature, 850 hPa temperature, upward longwave radiation, and downward longwave radiation. Daily variations of atmospheric variables are highly correlated with each other, suggesting that they have a common cause (see Figure R4). Specifically, the impact of synoptic disturbance is conspicuous with significant fluctuations on synoptic time scales.

Further, CSEOF analysis indicates that these variations are amplifying in time as reflected in the PC time series. The mechanism described in Figure 9 (old) is the winter average picture of the mechnism which is similar to that shown in Figure R3. We can average the CSLVs during winter to estimate the relative magnitude of change in heat flux or atmospheric variables as sea ice loss continues. Whatever the cause of sea ice reduction is, a 1% loss of sea ice results in the changes in other variables as described in Figure 9. Further, the lagged correlation analysis among these variables indicates that turbulent heat flux preceeds 850 hPa warming, which, in turn, is followed by increased downward longwave radiation (see Figure R4). Ultimately, surface air

temperature increases due to increased downward longwave radiation.

On the other hand, the mechanism addressed above cannot be demonstrated in CSEOF analysis of monthly data. The cause-and-effect relationship among the variables in Figure R4 can only be appreciated when we analyze 3-hourly data. Therefore, we remove the entire discussion associated with Figure 9. Hopefully, we will address this mechanism in a new paper where 3-hourly data is employed for CSEOF analysis. [Removed the text associated with Figure 9.]

Minor comments

C1: p. 1, l. 10: "Arctic" misspelled, article missing

R1: Thank you. Corrected. [P1 L10]

C2: p. 1, l. 14: remove "to be"

R2: Corrected. [P1 L14]

C3: p. 2, l. 4: "Serreze" misspelled

R3: Corrected. [P2 L5]

C4: p. 2, l. 7: "in the earlier period" please be specific

R4: The cited references differ in terms of "earlier" and "later" periods. Therefore, we cannot be specific about the definition of earlier period. [no corrective action]

C5. p. 2, l. 29: "remains to be melted" is awkward

R5: We rephrased it as "sea surface remains to be ice free". [P2 L29]

C6: p. 2, l. 32: "While summer sea ice melting is clearly seen. . ." Does this mean decreased summer sea ice concentrations? P. 3, l. 2: "winter sea ice melting" Is sea ice really melting during the winter? Please see general comment #1 above.

R6: We would like to keep the wording "sea ice melting" here, since it is sea ice melting.

In winter, however, it is not "melting" but "reduction". Therefore, we changed it to "sea ice loss in winter". [P3 L1]

C7: p. 3, l. 12-13: "each term in the feedback" is repeated

R7: We changed the sentence as follows: "... in order to clarify their relative importance in the feedback." [P3 L13]

C8: p. 3, l. 22: add "and" before "2 m temperature"

R8: Thank you. Complied. [P4 L4]

C9: p. 5, l. 4: "extract physically meaningful consistent evolutions from these variables" I was confused by this statement, perhaps because of the use of the word "evolutions"

R9: We used the word "evolution", since we are dealing with temporal variation of spatial patterns. We changed the wording as follows: "... extract physically consistent space-time evolution patterns from these variables." [P7 L5] We also changed the "method of analysis" section significantly so that the concept of CSEOF analysis can be more easily conveyed.

C10: p. 5, l. 9: "volatile" is not the word choice I would have expected

C10: We changed the word to "sensitive". [P7 L10]

C11: p. 6, l. 2: should be "increases" (agr)

R11: Thank you. Corrected. [P8 L5]

C12: p. 7, l. 21-22: "It is noted that" and "It is also worthy of remark that" are not necessary

R12: Complied. [P9 L26-28: Both the downward and upward radiation at the surface is maximized in winter (specifically February) with very small values in summer (Fig. 7b). Turbulent heat flux is maximized when 850 hPa temperature is minimum in March and November (Fig. 7c).]

C13: p. 8, l. 2: "is maintaining sea ice stay melted" is confusing

R13: We changed it as follows: "delayed warming is not so effective in sustaining the ice-free condition in winter in the ..." [P10 L6]

C14: p. 9, l. 21: "trapping" is not a good description of this process

R14: We changed "trapping" to "absorbing". [P11 L21]

C15: Figure 3 (and others): Some of the contours are difficult to follow, particularly on the JJA panel. If you chose not to label some of the contours, you may want to indicate the contour interval in the captions.

R15: We modified the contouring intervals to make the map readable. [See Figures 3-5 and Figure captions.]

* Figure Captions

Figure R1. The Daily patterns of variability over the region of sea ice loss ($21°$-$79.5°$E × $75°$-$79.5°$N): (a) sea ice concentration, (b) 850 hPa air temperature (black), turbulent flux (red), and specific humidity (blue)$×10$, and (c) specific humidity (blue) and moisture convergence (red).

Figure R2. The winter average pattern of moisture transport and convergence for the Arctic warming mode. This pattern is obtained by averaging daily patterns over DJF.

Figure R3. Daily patterns of variability over the region of sea ice loss ($21°$-$79.5°$E × $75°$-$79.5°$N): (a) sea ice concentration, (b) 2 m air temperature (red), 850 hPa air temperature$×2$ (black), and upward longwave radiation (blue), and (c) same as (b) except for the regressed downward longwave radiation (blue). Correlation of upward and downward longwave radiations with 2 m air temperature is respectively 0.90 and 0.95, whereas with 850 hPa air temperature is 0.60 and 0.86. (d) Corresponding PC time series.

Figure R4. Correlation of upward (solid lines) and downward (dotted lines) longwave

radiations with 2 m air temperature (blue), 850 hPa temperature (red), and sea ice concentration (black). Longwave radiation lags the other variable for a positive lag. Lagged correlation between 2 m air temperature and 850 hPa air temperature (black dashed line); 2 m air temperature leads 850 hPa temperature for a positive lag.

** The combined response file including a marked-up manuscript is attached.

Please also note the supplement to this comment:
http://www.the-cryosphere-discuss.net/tc-2016-69/tc-2016-69-AC3-supplement.pdf

[Figure]

**Fig. 1.** The Daily patterns of variability over the region of sea ice loss (21°-79.5°E × 75°-79.5°N): (a) sea ice concentration, (b) 850 hPa air temperature (black), turbulent flux (red), and specific humidity

[Figure]

**Fig. 2.** The winter average pattern of moisture transport and convergence for the Arctic warming mode. This pattern is obtained by averaging daily patterns over DJF.

[Figure]

**Fig. 3.** Daily patterns of variability over the region of sea ice loss (21°-79.5°E × 75°-79.5°N): (a) sea ice concentration, (b) 2 m air temperature (red), 850 hPa air temperature×2 (black), and upward longwav

[Figure]

**Fig. 4.** Correlation of upward (solid lines) and downward (dotted lines) longwave radiations with 2 m air temperature (blue), 850 hPa temperature (red), and sea ice concentration (black). Longwave radiation

**Supplement:**

The authors have examined the mechanisms by which declining sea ice in the Arctic is contributing to Arctic amplification of climate change. They focused on the differential changes in the Barents-Kara, Laptev and Chukchi seas and identified a unique pattern of change in the Barents and Kara seas associated with turbulent transport of heat from open water in the winter. The authors make a useful contribution to our understanding of the surface energy budget under conditions of

10   reduced sea ice in the Arctic.
I have a few general recommendations and some specific edits recommended:

General comments

15   Comment1(C1): The authors often talk about sea ice "melt" when they are referring to the trend toward reduced sea ice concentration. In some cases, "melt" may be the appropriate term, but in most cases it would be better to refer to reduced sea ice concentration and/or extent.

Response1(R1): Thank you for pointing this. We replaced "sea ice melting" by "sea ice loss" or "sea ice reduction" except
20   when we really meant "melting". [Corrections are scattered throughout the manuscript.]

C2: The authors discuss the increase in 850hPa temperatures over the Barents and Kara seas of less than 0.1K, which they indicate leads to a ~1 W/m2 in downwelling longwave, which leads to two questions (see p. 8, l. 25-34). a. How can the authors be certain that the change in 850hPa temperature is due to the reduced sea ice concentration? b. I did some back of
25   the envelope calculations, and it appears the magnitude of the downwelling longwave change is too large to be fully explained by the 850hPa temperature increase. Perhaps increased atmospheric moisture is playing a role in the increased downwelling longwave? The authors briefly discuss (p. 9, l. 20- 23) and cite the recent Park et al. paper (p. 12, l. 21-23), but perhaps this issue should be further explored/discussed.

30   R2a: As can be seen in Figure 10 (new Figure 9), the anomaly pattern of sea ice loss and those of turbulent heat flux, 2 m air temperature, upward longwave radiation, downward longwave radiation, and 850 hPa air temperature have common centers of action. These patterns share the same PC time series (Figure 2b). It seems reasonable to assume that these patterns share an identical source of variability. Figure R1 shows the loading vectors of the Arctic warming mode derived from the daily ERA-Interim data in winter (Dec. 1-Feb. 28; 90 days); the loading vectors are averaged over the Barents-Kara Seas. As can

be seen in the figure, sea ice reduction is clearly seen throughout the winter (Figure R1a). The loading vector of turbulent heat flux has a positive mean and is generally positive throughout the winter because of the sea ice reduction in the area (Figure R1b; red curve)). The 850 hPa air temperature anomaly is also positive with a mean value of ~1.26 K (Figure R1b; black curve). Finally, the loading vector of specific humidity has a mean value of ~0.15 g kg$^{-1}$, and is highly correlated with

5 the 850 hPa air temperature; correlation is 0.91 (Figure R1b; blue curve). The specific humidity is moderately correlated with moisture convergence (corr = 0.54). Turbulent heat flux is negatively correlated with 850 hPa air temperature and specific humidity. This negative correlation seems to indicate that turbulent heat flux decreases as air temperature increases and specific humidity increases and reflects the bulk formulas for latent and sensible heat flux. It should be noted that all the variables have significant positive means. This positive mean is due to sea ice reduction throughout the winter. On top of

10 the increased turbulent

[Figure]

**Figure R1**. The Daily patterns of variability over the region of sea ice loss (21°-79.5°E × 75°-79.5°N): (a) sea ice concentration, (b) 850 hPa air temperature (black), turbulent flux (red), and specific humidity (blue) ×10, and (c) specific humidity (blue) and moisture convergence (red).

[Figure]

**Figure R2**. The winter average pattern of moisture transport and convergence for the Arctic warming mode. This pattern is obtained by averaging daily patterns over DJF.

heat flux through the open surface of the ocean, the release of the turbulent heat flux is affected by the atmospheric condition (such as air temperature and humidity) as well as horizontal moisture transport.

Figure R2 shows the winter average pattern of moisture transport and convergence for the Arctic warming mode. This pattern is similar to Figure 5 in Park et al. (2015). There is a sign of moisture convergence in the Barents-Kara Seas. On the other hand, the daily time series over the Barents-Kara Seas (Figure R1c) indicates that the sign of moisture convergence fluctuates around zero. Thus, it is difficult to explain the non-zero mean of specific humidity in Figure R1b in terms of the moisture transport and convergence.

Thus, we think that the loss of sea ice leads to increased turbulent heat flux, which not only warms the atmospheric column but also increases specific humidity. Saturation specific humidity also increases as the atmospheric column warms up. The increased air temperature and specific humidity both contribute to the increased downward longwave radiation. This discussion is difficult to include in the revision, since it requires new analysis based on daily ERA-Interim reanalysis data. [no corrective action]

R2b: According to the first law of thermodynamics, we have

$$\rho c_p \frac{\partial T}{\partial t} = -\nabla \cdot \vec{F},$$
(1)

where $\rho$ is the density of air, $c_p$ is specific heat at constant pressure, $T$ is temperature, and $\vec{F}$ is heat flux. In a one-dimensional column model, (1) can be rewritten as

$$\rho c_p \frac{\partial T}{\partial t} = -\frac{\partial}{\partial z}\left(F^\uparrow - F^\downarrow\right) = -\frac{\partial}{\partial z}\left(F_{net}^\uparrow\right).$$
(2)

By integrating (2) with respect $z$ from level $z_1$ to $z_2$, we have

$$\rho c_p \frac{\partial}{\partial t}\left(\int_{z_1}^{z_2} T\, dz\right) = -F_{net}^\uparrow \Big|_{z_1}^{z_2}.$$
(3)

Let $z_1 = \varepsilon$ is the level at which radiative transfer is zero (say, slightly below the surface). Then, we can show that

$$\rho c_p \frac{\partial}{\partial t}\left(\int_{\varepsilon}^{z_2} T\, dz\right) = -F_{net}^\uparrow(z = z_2) = F^\downarrow(z_2) - F^\uparrow(z_2).$$
(4)

If $z_2$ represents a vertical level slightly above the surface, the temperature near the surface is determined by downward and upward flux at level $z = z_2$. According to Figure 9, downward flux is larger than upward flux at the surface. Henceforth, surface temperature should increase. Likewise, we can let $z_2$ be the 850 hPa level and determine net heat flux for a temperature change integrated from 850 hPa to surface. Nonetheless, we cannot calculate upward flux and downward flux separately; we can only calculate net flux. Therefore, we cannot show what the downward flux from the 850 hPa level. In short, it is not the downward flux but the net flux that is related to the temperature change at the 850 hPa level (0.07 K). Of course, we do not have flux information at all vertical levels and we cannot verify theoretically that the downward flux 0.97 W m$^{-2}$ is due to temperature change. [We eliminated Figure 9 together with the text associated with it.]

C3: I appreciated that the authors added a schematic to explain the processes involved in the Barents and Kara seas (Fig. 9). Unfortunately, I was still left somewhat confused by the figure. For example, the "Increase T 0.07K" does not indicate at what level. The arrows leave me wondering if this is all happening concurrently or if there is a time associated with each process. I think this schematic is a good idea but could benefit from some additional thought.

R3: As explained in the method section, regression in CSEOF space allows us to write data in the form:

$$Data(r,t) = \sum_n \left\{ B_n(r,t), C_n^{(r)}(r,t), D_n^{(r)}(r,t), E_n^{(r)}(r,t), \cdots \right\} T_n(t),$$

where the terms in curly braces for each $n$ represents a physical process as reflected in different variables (say, temperature, sea ice concentration, 850 hPa air temperature, upward longwave radiation, etc.). The terms in curly braces are physically consistent with each other. For example, Figure R3 below shows the daily evolution associated with the Arctic Amplification mode. CSEOF analysis was conducted on the daily ERA-Interim data during winter (Dec. 1-Feb. 28), and the
5   first CSEOF mode represents Arctic Amplification as in the present analysis. Shown in Figure R3 are the terms in curly braces for five different variables averaged over the Barents-Kara Seas [21°-79.5°E × 75°-79.5°N]. As can be seen in the figure, loss of sea ice is reflected in the positive values of anomalous 2 m air temperature, 850 hPa temperature, upward longwave radiation, and downward longwave radiation. Daily variations of atmospheric variables are highly correlated with each other, suggesting that they have a common cause (see Figure R4). Specifically, the impact of synoptic disturbance is
10  conspicuous with significant fluctuations on synoptic time scales.

Further, CSEOF analysis indicates that these variations are amplifying in time as reflected in the PC time series. The mechanism described in Figure 9 (old) is the winter average picture of the mechnism which is similar to that shown in Figure R3. We can average the CSLVs during winter to estimate the relative magnitude of change in heat flux or atmospheric
15  variables as sea ice loss continues. Whatever the cause of sea ice reduction is, a 1% loss of sea ice results in the changes in other variables as described in Figure 9. Further, the lagged correlation analysis among these variables indicates that turbulent heat flux preceeds 850 hPa warming, which, in turn, is followed by increased downward longwave radiation (see Figure R4). Ultimately, surface air temperature increases due to increased downward longwave radiation.

20  On the other hand, the mechanism addressed above cannot be demonstrated in CSEOF analysis of monthly data. The cause-and-effect relationship among the variables in Figure R4 can only be appreciated when we analyze 3-hourly data. Therefore, we remove the entire discussion associated with Figure 9. Hopefully, we will address this mechanism in a new paper where 3-hourly data is employed for CSEOF analysis. [Removed the text associated with Figure 9.]

[Figure]

**Figure R3**. Daily patterns of variability over the region of sea ice loss (21°-79.5°E × 75°-79.5°N): (a) sea ice concentration, (b) 2 m air temperature (red), 850 hPa air temperature × 2 (black), and upward longwave radiation (blue), and (c) same as (b) except for the regressed downward longwave radiation (blue). Correlation of upward and downward longwave radiations with 2 m air temperature is respectively 0.90 and 0.95, whereas with 850 hPa air temperature is 0.60 and 0.86. (d) Corresponding PC time series.

[Figure]

**Figure R4**. Correlation of upward (solid lines) and downward (dotted lines) longwave radiations with 2 m air temperature (blue), 850 hPa temperature (red), and sea ice concentration (black). Longwave radiation lags the other variable for a positive lag. Lagged correlation between 2 m air temperature and 850 hPa air temperature (black dashed line); 2 m air temperature leads 850 hPa temperature for a positive lag.

Minor comments

C1: p. 1, l. 10: "Arctic" misspelled, article missing

R1: Thank you. Corrected. [P1 L10]

C2: p. 1, l. 14: remove "to be"

R2: Corrected. [P1 L14]

C3: p. 2, l. 4: "Serreze" misspelled

R3: Corrected. [P2 L5]

C4: p. 2, l. 7: "in the earlier period" please be specific

R4: The cited references differ in terms of "earlier" and "later" periods. Therefore, we cannot be specific about the definition of earlier period. [no corrective action]

C5. p. 2, l. 29: "remains to be melted" is awkward

R5: We rephrased it as "sea surface remains to be ice free".  [P2 L29]

5   C6: p. 2, l. 32: "While summer sea ice melting is clearly seen. . ." Does this mean decreased summer sea ice concentrations?
P. 3, l. 2: "winter sea ice melting" Is sea ice really melting during the winter? Please see general comment #1 above.

R6: We would like to keep the wording "sea ice melting" here, since it is sea ice melting.  In winter, however, it is not
"melting" but "reduction".  Therefore, we changed it to "sea ice loss in winter".  [P3 L1]

C7: p. 3, l. 12-13: "each term in the feedback" is repeated

R7: We changed the sentence as follows: "… in order to clarify their relative importance in the feedback."  [P3 L13]

15   C8: p. 3, l. 22: add "and" before "2 m temperature"

R8: Thank you.  Complied.  [P4 L4]

C9: p. 5, l. 4: "extract physically meaningful consistent evolutions from these variables" I was confused by this statement,
20   perhaps because of the use of the word "evolutions"

R9: We used the word "evolution", since we are dealing with temporal variation of spatial patterns.  We changed the
wording as follows: "… extract physically consistent space-time evolution patterns from these variables."  [P7 L5]
We also changed the "method of analysis" section significantly so that the concept of CSEOF analysis can be more easily
25   conveyed.

C10: p. 5, l. 9: "volatile" is not the word choice I would have expected

C10: We changed the word to "sensitive".  [P7 L10]

C11: p. 6, l. 2: should be "increases" (agr)

R11: Thank you.  Corrected.  [P8 L5]

R12: Complied. [P9 L26-28: Both the downward and upward radiation at the surface is maximized in winter (specifically February) with very small values in summer (Fig. 7b). Turbulent heat flux is maximized when 850 hPa temperature is minimum in March and November (Fig. 7c).]

R13: We changed it as follows: "delayed warming is not so effective in sustaining the ice-free condition in winter in the …" [P10 L6]

R14: We changed "trapping" to "absorbing". [P11 L21]

R15: We modified the contouring intervals to make the map readable. [See Figures 3-5 and Figure captions.]

**Mechanism of Seasonal Arctic Sea Ice Evolution and Arctic Amplification**

Kwang-Yul Kim[1][*], Benjamin D. Hamlington[2], Hanna Na[3], and Jinju Kim[1]

[revised manuscript text omitted]

Kwang Yul Kim 6/27/2016 9:41 AM

Incr
0.

Increased LW↓    Sea ice
0.97 W m⁻²

2 m T
0.

Kwang Yul Kim 6/27/2016 9:42 AM